# Beyond Benign Overfitting in Nadaraya-Watson Interpolators

**Daniel Barzilai**
Weizmann Institute of Science
daniel.barzilai@weizmann.ac.il

**Guy Kornowski**
Weizmann Institute of Science
guy.kornowski@weizmann.ac.il

**Ohad Shamir**
Weizmann Institute of Science
ohad.shamir@weizmann.ac.il

## Abstract

In recent years, there has been much interest in understanding the generalization behavior of interpolating predictors, which overfit on noisy training data. Whereas standard analyses are concerned with whether a method is consistent or not, recent observations have shown that even inconsistent predictors can generalize well. In this work, we revisit the classic interpolating Nadaraya-Watson (NW) estimator (also known as Shepard's method), and study its generalization capabilities through this modern viewpoint. In particular, by varying a single bandwidth-like hyperparameter, we prove the existence of multiple overfitting behaviors, ranging non-monotonically from catastrophic, through benign, to tempered. Our results highlight how even classical interpolating methods can exhibit intricate generalization behaviors. In addition, for the purpose of tuning the hyperparameter, the results suggest that over-estimating the intrinsic dimension of the data is less harmful than under-estimating it. Numerical experiments complement our theory, demonstrating the same phenomena.

## 1 Introduction

The incredible success of over-parameterized machine learning models has spurred a substantial body of work, aimed at understanding the generalization behavior of interpolating methods (which perfectly fit the training data). In particular, according to classical statistical analyses, interpolating inherently noisy training data can be harmful in terms of test error, due to the bias-variance tradeoff. However, contemporary interpolating methods seem to defy this common wisdom [1, 2]. Therefore, a current fundamental question in statistical learning is to understand when models that perfectly fit noisy training data can still achieve strong generalization performance.

The notion of what it means to generalize well has somewhat changed over the years. Classical analysis has been mostly concerned with whether or not a method is consistent, meaning that asymptotically (as the training set size increases), the excess risk converges to zero. By now, several settings have been identified where even interpolating models may be consistent, a phenomenon known as "benign overfitting" [3–6]. However, following Mallinar et al. [7], a more nuanced view of overfitting has emerged, based on the observation that not all inconsistent learning rules are necessarily unsatisfactory.

In particular, it has been argued both empirically and theoretically that in many realistic settings, benign overfitting may not occur, yet interpolating methods may still overfit in a "tempered" manner, meaning that their excess risk is proportional to the Bayes error. On the other hand, in some situations

39th Conference on Neural Information Processing Systems (NeurIPS 2025).

overfitting may indeed be "catastrophic", leading to substantial degradation in performance even in the presence of very little noise. The difference between these regimes is significant when the amount of noise in the data is relatively small, and in such a case, models that overfit in a tempered manner may still generalize relatively well, while catastrophic methods do not. These observations led to several recent works aiming at characterizing which overfitting profiles occur in different settings beyond consistency, mostly for kernel regression and shallow ReLU networks [8–14]. We note that one classical example of tempered overfitting is 1-nearest neighbor, which asymptotically achieves at most *twice* the Bayes error [15]. Moreover, results of a similar flavor are known for $k$-nearest neighbor where $k > 1$ (see [16]). However, unlike the interpolating predictors we study here, $k$-nearest neighbors do not necessarily interpolate the training data when $k > 1$.

With this modern nuanced approach in mind, we revisit in this work one of the earliest and most classical learning rules, namely the Nadaraya-Watson (NW) estimator [17, 18]. In line with recent analysis focusing on interpolating predictors, we focus on an interpolating variant of the NW estimator, for binary classification: given (possibly noisy) classification data $S = (\mathbf{x}_i, y_i)_{i=1}^m \subset \mathbb{R}^d \times \{\pm 1\}$ sampled from some continuous distribution $\mathcal{D}$, and given some $\beta > 0$, we consider the predictor

$$\hat{h}_\beta(\mathbf{x}) := \begin{cases} \text{sign}\left(\sum_{i=1}^m \frac{y_i}{\|\mathbf{x} - \mathbf{x}_i\|^\beta}\right) & \text{if } \mathbf{x} \notin S \\ y_i & \text{if } \mathbf{x} = \mathbf{x}_i \text{ for some } \mathbf{x}_i \in S . \end{cases} \tag{1}$$

The predictor in Eq. (1) has a long history in the literature and is known by many different names, such as Shepard's method, inverse distance weighting (IDW), the Hilbert kernel estimate, and singular kernel classification (see Section 2 for a full discussion).

Notably, for *any* choice of $\beta > 0$, $\hat{h}_\beta$ interpolates the training set, meaning that $\hat{h}_\beta(\mathbf{x}_i) = y_i$. We will study the predictor's generalization in "noisy" classification tasks: we assume there exists a ground truth $f^* : \mathbb{R}^d \to \{\pm 1\}$ (satisfying mild regularity assumptions), so that for each sampled point $\mathbf{x}$, its associated label $y \in \{\pm 1\}$ satisfies $\Pr[y = f^*(\mathbf{x}) \,|\, \mathbf{x}] = 1 - p$ for some $p \in (0, 0.49)$. Clearly, for this distribution, no predictor can achieve expected classification error better than $p > 0$. However, interpolating predictors achieve $0$ training error on the training set, and thus by definition overfit. We are interested in studying the ability of these predictors to achieve low classification error with respect to the underlying distribution. Factoring out the inevitable error due to noise, we can measure this via the "clean" classification error $\Pr_{\mathbf{x} \sim \mathcal{D}_\mathbf{x}}[\hat{h}_\beta(\mathbf{x}) \neq f^*(\mathbf{x})]$, which measures how well $\hat{h}_\beta$ captures the ground truth function $f^*$.

As our starting point, we recall that $\hat{h}_\beta$ is known to exhibit benign overfitting when $\beta = d$ precisely:

**Theorem 1.1** (Devroye et al. [19]). *Suppose $\mathcal{D}_\mathbf{x}$ has a density on $\mathbb{R}^d$, and let $\beta = d$. For any noise level $p \in (0, 0.49)$, it holds that the clean classification error of $\hat{h}_\beta$ goes to zero as $m \to \infty$, i.e. $\hat{h}_\beta$ exhibits benign overfitting.*

In other words, although training labels are flipped with probability $p \in (0, 0.49)$, the predictor is asymptotically consistent, and thus predicts according to the ground truth $f^*$. Furthermore, Devroye et al. [19] also informally argued that setting $\beta \neq d$ is inconsistent in general, and therefore excess risk should be expected. Nonetheless, the behavior of the predictor $\hat{h}_\beta$ beyond the benign/consistent setting is not known prior to this work.

In this paper, in light of the recent interest in inconsistent interpolation methods, we characterize the price of overfitting in the inconsistent regime $\beta \neq d$. What is the nature of the inconsistency for $\beta \neq d$? Is the overfitting tempered, or in fact catastrophic? As our main contribution, we answer these questions and prove the following asymmetric behavior:

**Theorem 1.2** (Main results, informal). *For any dimension $d \in \mathbb{N}$ and noise level $p \in (0, 0.49)$, the following hold asymptotically as $m \to \infty$:*

- *("Tempered" overfitting) For any $\beta > d$, the clean classification error of $\hat{h}_\beta$ is between $\Omega(\text{poly}(p))$ and $\widetilde{\mathcal{O}}(p)$.*

- *("Catastrophic" overfitting) For any $\beta < d$, there is some $f^*$ for which $\hat{h}_\beta$ will suffer constant clean classification error, independently of $p$.*

We summarize the overfitting profile that unfolds in Figure 1, with an illustration of the Nadaraya-Watson interpolator in one dimension. These results provide a modern analysis of a classical learning

rule, uncovering a range of generalization behaviors: By varying a single hyperparameter, these behaviors range non-monotonically from catastrophic to tempered overfitting, with a delicate sliver of benign overfitting behavior in between. Our results highlight how intricate generalization behaviors, including the full range from benign to catastrophic overfitting, can appear in simple and well-known interpolating learning rules. To the best of our knowledge, for kernel interpolators, there is no other example of a single kernel provably exhibiting all three types of overfitting as we do here (even with a varying bandwidth).

Moreover, the results provide an interesting insight about the optimal tuning of $\beta$: Although Theorem 1.1 might seem to suggest that the optimal value of $\beta$ is simply the input dimension $d$, it does not cover the common case where the data has some intrinsic dimension $d_{\text{int}} < d$ (due to the requirement that $\mathcal{D}_{\mathbf{x}}$ has a density on $\mathbb{R}^d$). In that situation, our analysis suggests that the optimal value for $\beta$ is in fact $d_{\text{int}}$, not $d$. Unfortunately, $d_{\text{int}}$ is generally not known, and can only be estimated. In that case, our results suggest that setting $\beta$ to an *over-estimate* of $d_{\text{int}}$ (namely, choosing some $\beta > d_{\text{int}}$) is much preferable to under-estimating it, as the former leads to tempered overfitting, whereas the latter may lead to catastrophic overfitting. We further discuss this in Remark 5.2, and in Section 6 we present numerical evidence supporting this claim.

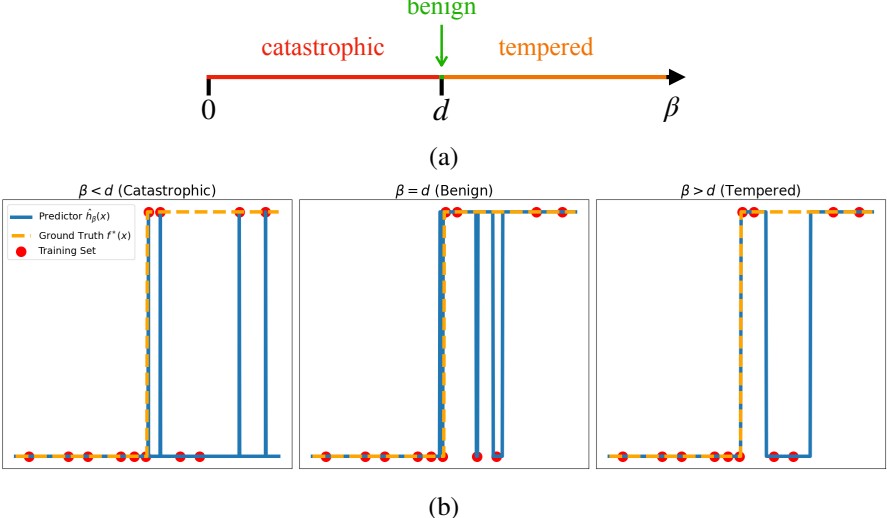

(a)

(b)

Figure 1: (a): Illustration of the entire overfitting profile of the NW interpolator given by Eq. (1). (b): Toy illustration of the NW interpolator in dimension $d = 1$ with noisy data. **(Left)** Catastrophic overfitting for $\beta < d$: the prediction at each point is influenced too heavily by far-away points, and therefore the predictor does not capture the general structure of the ground truth function $f^*$. **(Middle)** Benign overfitting for $\beta = d$: asymptotically the excess risk will be Bayes-optimal. **(Right)** Tempered overfitting for $\beta > d$, the prediction at each point is influenced too heavily by nearby points, so the predictor misclassifies large regions around label-flipped points, but only around them.

The paper is structured as follows. In Section 2, we review related work. In Section 3 we formally present the discussed setting. In Section 4, we present our result for the tempered regime $\beta > d$. In Section 5 we present our result for the catastrophic regime $\beta < d$. In Section 6 we provide some illustrative experiments to complement our theoretical findings. We conclude in Section 7. All of the results in the main text include proof sketches, while full proofs appear in the appendix.

## 2   Related work

**Nadaraya-Watson kernel estimator.**     The Nadaraya-Watson (NW) estimator was introduced independently in the seminal works of Nadaraya [17] and Watson [18]. Later, and again independently, in the context of reconstructing smooth surfaces, Shepard [20] used a method referred to as Inverse Distance Weighting (IDW), which is in fact a NW estimator with respect to certain kernels leading to interpolation, identical to those we consider in this work. To the best of our knowledge, Devroye et al. [19] provided the first statistical guarantees for such interpolating NW estimators (which they

called the Hilbert kernel), showing that the predictor given by Eq. (1) with $\beta = d$ is asymptotically consistent. For a more general discussion on so called "kernel rules", see [16, Chapter 10]. In more recent works, Belkin et al. [21] derived non-asymptotic rates showing consistency under a slight variation of the kernel. Radhakrishnan et al. [22], Eilers et al. [23] showed that in certain cases, neural networks in the NTK regime behave approximately as the NW estimator, and leverage this to show consistency. Abedsoltan et al. [24] showed that interpolating NW estimators can be used in a way that enables in-context learning.

**Overfitting and generalization.** There is a substantial body of work aimed at analyzing the generalization properties of interpolating predictors that overfit noisy training data. Many works study settings in which interpolating predictors exhibit benign overfitting, such as linear predictors [3, 25–30], kernel methods [31, 32, 6], and other learning rules [19, 33, 1].

On the other hand, there is also a notable line of work studying the limitations of generalization bounds in interpolating regimes [34, 2, 35]. In particular, several works showed that various kernel interpolating methods are not consistent in any fixed dimension [36–38], or whenever the number of samples scales as an integer-degree polynomial with the dimension [39, 40, 12, 41].

Motivated by these results and by additional empirical evidence, Mallinar et al. [7] proposed a more nuanced view of interpolating predictors, coining the term *tempered overfitting* to refer to settings in which the asymptotic risk is strictly worse than optimal, but is still better than a random guess. A well-known example is the classic 1-nearest-neighbor interpolating method, for which the excess risk scales linearly with the probability of a label flip [15]. Several works subsequently studied settings in which tempered overfitting occurs in the context of kernel methods [11, 12, 42], and for other interpolation rules [8, 9, 43].

Finally, some works studied settings in which interpolating with kernels is in fact *catastrophic*, meaning that the excess error is lower bounded by a constant which is independent of the noise level, leading to substantial risk even in the presence of very little noise [9, 10, 13, 14].

We note that our proof techniques differ from most known results for kernel interpolators, which typically rely on a spectral analysis. However, this often requires additional non-trivial assumptions (e.g. Gaussian universality). By contrast, our proofs are based on characterizing the "locality" of the predictor.

**Varying kernel bandwidth.** Several works considered generalization bounds that hold uniformly over a family of kernels, parameterized by a bandwidth parameter [36, 44, 37, 38, 13]. The bandwidth plays the same role as the parameter $\beta$ in this paper, controlling how local/global the kernel is. Specifically, these works showed that in fixed dimensions various kernels are asymptotically inconsistent *for all* bandwidths. Medvedev et al. [13] showed that with large enough noise, the Gaussian kernel with any bandwidth is at least as bad as a constant predictor, which we classify as catastrophic. As far as we know, our paper provides the first known example of a kernel method provably exhibiting all types of overfitting behaviors in fixed dimensions by varying the bandwidth alone.

## 3 Preliminaries

**Notation.** We use bold-faced font to denote vectors, e.g. $\mathbf{x} \in \mathbb{R}^d$, and denote by $\|\mathbf{x}\|$ the Euclidean norm. We let $[n] := \{1, \ldots, n\}$. Given some set $A \subseteq \mathbb{R}^d$ and a function $f$, we denote its restriction by $f|_A : A \to \mathbb{R}$, and by $\mathrm{Unif}(A)$ the uniform distribution over $A$. We let $B(\mathbf{x}, r) := \{\mathbf{z} \mid \|\mathbf{x} - \mathbf{z}\| \leq r\}$ be the ball of radius $r$ centered at $\mathbf{x}$. We denote by $\overset{d}{=}$ equality in distribution. We use the standard big-O notation, with $\mathcal{O}(\cdot)$, $\Theta(\cdot)$ and $\Omega(\cdot)$ hiding absolute constants that do not depend on problem parameters, and $\tilde{\mathcal{O}}(\cdot)$, $\tilde{\Omega}(\cdot)$ additionally hiding logarithmic factors. Given some parameter (or set of parameters) $\theta$, we denote by $c(\theta), C(\theta), C_1(\theta), \widetilde{C}(\theta)$ etc. positive constants that depend on $\theta$.

**Setting.** Given some target function $f^* : \mathbb{R}^d \to \{\pm 1\}$, we consider a classification task based on noisy training data $S = (\mathbf{x}_i, y_i)_{i=1}^m \subset \mathbb{R}^d \times \{\pm 1\}$, such that $\mathbf{x}_1, \ldots, \mathbf{x}_m \sim \mathcal{D}_{\mathbf{x}}$ are sampled from some distribution $\mathcal{D}_{\mathbf{x}}$ with a density $\mu$, and for each $i \in [m]$ independently, $y_i = f^*(\mathbf{x}_i)$ with probability $1 - p$ or else $y_i = -f^*(\mathbf{x}_i)$ with probability $p \in (0, 0.49)$. We note that while we focus

on a fixed noise level $p$ for simplicity, our results can also be extended to the case where $p$ varies smoothly with $\mathbf{x}$.

Given the predictor $\hat{h}_\beta$ introduced in Eq. (1), we denote the asymptotic clean classification error by[1]

$$\mathcal{L}(\hat{h}_\beta) = \lim_{m\to\infty} \mathbb{E}_S\left[\Pr_{\mathbf{x}\sim\mathcal{D}_\mathbf{x}}[\hat{h}_\beta(\mathbf{x}) \neq f^*(\mathbf{x})]\right].$$

Throughout the paper we impose the following mild regularity assumptions on $\mu$ and $f^*$ :

**Assumption 3.1.** *We assume $\mu$ is continuous at almost every $\mathbf{x} \in \mathbb{R}^d$. We also assume that for almost every $\mathbf{x} \in \mathbb{R}^d$, there is a neighborhood $B_\mathbf{x} \supset \{\mathbf{x}\}$ such that $f^*|_{B_\mathbf{x}} \equiv f^*(\mathbf{x})$.*

We note that the assumptions above are very mild. Indeed, any density is Lebesgue integrable, whereas our assumption for $\mu$ is equivalent to it being Riemann integrable. As for $f^*$, the assumption asserts that its associated decision boundary has zero measure, ruling out pathological functions.

**Types of overfitting.** We study the asymptotic error guaranteed by $\hat{h}_\beta$ in a "minimax" sense, namely uniformly over $\mu, f^*$ that satisfy Assumption 3.1. Under the described setting with noise level $p \in (0, 0.49)$, we say that:

- $\hat{h}_\beta$ exhibits benign overfitting if $\mathcal{L}(\hat{h}_\beta) = 0$;

- Else, $\hat{h}_\beta$ exhibits tempered overfitting if $\mathcal{L}(\hat{h}_\beta)$ scales monotonically with $p$: there exists $\varphi : [0,1] \to [0,1]$ non-decreasing, continuous with $\varphi(0) = 0$, so that $\mathcal{L}(\hat{h}_\beta) \leq \varphi(p)$;

- $\hat{h}_\beta$ exhibits catastrophic overfitting if there exist some $\mu, f^*$ (satisfying the regularity assumptions) such that $\mathcal{L}(\hat{h}_\beta)$ is lower bounded by a positive constant (independent of $p$).

We remark that the latter definition of catastrophic overfitting slightly differs from the one of Mallinar et al. [7], which called the method catastrophic only if $\mathcal{L}(\hat{h}_\beta) = \frac{1}{2}$. Medvedev et al. [13] noted that the latter definition can result in even the most trivial predictor, a function that is constant outside the training set, being classified as tempered instead of catastrophic. We therefore find the formalization above more suitable, which also coincides with previous works [8, 9, 12, 13, 43].

## 4   Tempered overfitting

We start by presenting our main result for the $\beta > d$ parameter regime, establishing tempered overfitting of the predictor $\hat{h}_\beta$ :

**Theorem 4.1.** *For any $d \in \mathbb{N}$, any $\beta > d$, any density $\mu$ and target function $f^*$ satisfying Assumption 3.1, and any noise level $p \in (0, 0.49)$, it holds that*

$$C_1(\beta/d) \cdot p^{c(\beta/d)} \;\leq\; \mathcal{L}(\hat{h}_\beta) \;\leq\; C_2(\beta/d) \cdot \log^{\frac{1}{1-d/\beta}}(1/p) \cdot p,$$

*where $c(\beta/d) = \left(\frac{8\cdot 2^{\beta/d}}{\beta/d-1}\right)^{\frac{1}{\beta/d-1}} > 0$, and $C_1(\beta/d), C_2(\beta/d) > 0$ are constants that depend only on the ratio $\beta/d$.*

In particular, the theorem implies that for any $\beta > d$ it holds that $\mathcal{L}(\hat{h}_\beta) = \widetilde{\mathcal{O}}(p)$, hence in low noise regimes the error is never too large. Moreover, we note that the lower bound (of the form $\Omega(\mathrm{poly}(p))$ for any $\beta > d$) holds for *any* target function satisfying mild regularity assumptions. Therefore, the tempered cost of overfitting holds not only in a minimax sense, but for any instance.

Further note that since we know that $\beta = d$ leads to benign overfitting, one should expect the lower bound in Theorem 4.1 to approach 0 as $\beta \to d^+$. Indeed, the lower bound's polynomial degree satisfies $c(\beta/d) = \left(\frac{8\cdot 2^{\beta/d}}{\beta/d-1}\right)^{\frac{1}{\beta/d-1}} \overset{\beta\to d^+}{\longrightarrow} \infty$, and thus $p^{c(\beta/d)} \overset{\beta\to d^+}{\longrightarrow} 0$.[2]

---

[1]Technically, the limit may not exist in general. In that case, our lower bounds hold for the $\liminf_{m\to\infty}$, while our upper bounds hold for the $\limsup_{m\to\infty}$, and therefore both hold for all partial limits.

[2]To be precise, one needs to make sure that the constant $C_1(\beta/d)$ does not blow up, which is indeed the case.

We provide below a sketch of the main ideas that appear in the proof of Theorem 4.1, which is provided in Appendix B. In a nutshell, the proof establishes that when $\beta > d$, the predictor $\hat{h}_\beta$ is highly *local*, and thus prediction at a test point is affected by flipped labels nearby, yet only by them. The proof essentially shows that in this parameter regime, $\hat{h}_\beta$ behaves similar to the $k$ nearest neighbor ($k$-NN) method for some finite $k$ that depends on $\beta/d$ (although notably, as opposed to $\hat{h}_\beta$, $k$-NN does not interpolate), and has a similarly tempered generalization guarantee accordingly.

*Proof sketch of Theorem 4.1.* Looking at some test point $\mathbf{x} \in \mathbb{R}^d$, we are interested in understanding the prediction $\hat{h}_\beta(\mathbf{x})$. Clearly, by definition in Eq. (1), the prediction depends on the random variables $\|\mathbf{x} - \mathbf{x}_i\|^{-\beta}$ for $i \in [m]$, so that closer datapoints have a great affect on the prediction at $\mathbf{x}$. Denote by $y_{(1)}, \ldots, y_{(m)}$ the labels ordered according to the distance of their corresponding datapoints, namely $\|\mathbf{x} - \mathbf{x}_{(1)}\| \leq \|\mathbf{x} - \mathbf{x}_{(2)}\| \leq \cdots \leq \|\mathbf{x} - \mathbf{x}_{(m)}\|$. By analyzing the distribution of distances from the sample to $\mathbf{x}$, for datapoints sufficiently close to $\mathbf{x}$ we can jointly approximate the random variables by $\|\mathbf{x} - \mathbf{x}_{(i)}\|^{-\beta} \approx \mu(\mathbf{x})(\sum_{i=1}^{m+1} E_i)^{\beta/d}/(\sum_{j=1}^{i} E_j)^{\beta/d}$, where $E_1, \ldots, E_m \overset{i.i.d.}{\sim} \exp(1)$ are standard exponential random variables. Furthermore, datapoints which are more than some constant distance away from $\mathbf{x}$ can contribute at most a constant, so for some $m' < m$ we obtain

$$\hat{h}_\beta(\mathbf{x}) \approx \text{sign}\left( \mu(\mathbf{x})\left( \sum_{i=1}^{m+1} E_i \right)^{\beta/d} \sum_{i=1}^{m'} \frac{y_{(i)}}{(\sum_{j=1}^{i} E_j)^{\beta/d}} + \mathcal{O}(m) \right). \tag{2}$$

Since $\mathbb{E}[\sum_{j=1}^{i} E_j] = i$, we apply concentration bounds for sums of exponential variables to argue that with high probability $\sum_{j=1}^{i} E_j \approx i$ simultaneously over all $i \in \mathbb{N}$, so the prediction is roughly

$$\hat{h}_\beta(\mathbf{x}) \approx \text{sign}\left( \mu(\mathbf{x})(m+1)^{\beta/d} \sum_{i=1}^{m'} \frac{y_{(i)}}{i^{\beta/d}} + \mathcal{O}(m) \right) \approx \text{sign}\left( \sum_{i=1}^{m'} \frac{y_{(i)}}{i^{\beta/d}} \right),$$

since $\mathcal{O}(m) \ll (m+1)^{\beta/d}$ is asymptotically negligible and $\mu(\mathbf{x})(m+1)^{\beta/d} > 0$.

Crucially, for any $\beta > d$, the sum above converges, and therefore there exists a constant $k \in \mathbb{N}$ (that depends only on the ratio $\beta/d$) so that the tail is smaller than the first $k$ summands:

$$\left| \sum_{i=k+1}^{m'} \frac{y_{(i)}}{i^{\beta/d}} \right| \lesssim \sum_{i=k+1}^{\infty} \frac{1}{i^{\beta/d}} \lesssim \frac{1}{k^{\beta/d-1}} \ll \sum_{i=1}^{k} \frac{1}{i^{\beta/d}} .$$

Therefore, under the event that all nearby labels coincide, the prediction depends only on the $k$ nearest neighbors, and we would get that predictor returns their value. By Assumption 3.1, for sufficiently large sample size $m$ and fixed $k$, for almost every $\mathbf{x}$ the $k$ nearest neighbors should be labeled the same as $\mathbf{x}$, namely $f^*(\mathbf{x}) = f^*(\mathbf{x}_{(1)}) = \cdots = f^*(\mathbf{x}_{(k)})$. So overall, we see that

$$\Pr[\hat{h}_\beta(\mathbf{x}) \neq f^*(\mathbf{x})] \leq \Pr[\underbrace{\exists i \in [k]: \ y_{(i)} \neq f^*(\mathbf{x}_{(i)})}_{\text{flipped label}}] = 1 - (1-p)^k \leq kp ,$$

and similarly

$$\Pr[\hat{h}_\beta(\mathbf{x}) \neq f^*(\mathbf{x})] \geq \Pr[\underbrace{\forall i \in [k]: \ y_{(i)} \neq f^*(\mathbf{x}_{(i)})}_{\text{all } k \text{ labels flipped}}] = p^k .$$

The two inequalities above show the desired upper and lower bounds on the prediction error.

$\square$

## 5 Catastrophic overfitting

We now turn to present our main result for the $\beta < d$ parameter regime, establishing that $\hat{h}_\beta$ can catastrophically overfit:

**Theorem 5.1.** *For any $d \in \mathbb{N}$ and any $0 < \beta < d$, there exist a density $\mu$ and a target function $f^*$ satisfying Assumption 3.1, such that for some absolute constants $C_1, C_2 \in (0,1)$, and $c(\beta, d) := C_1^{\beta} \cdot (1 - \beta/d) > 0$, it holds for any $p \in (0, 0.49)$ that*

$$\mathcal{L}(\hat{h}_\beta) \geq C_2 \cdot c(\beta, d) \ .$$

The theorem states that whenever $\beta < d$, the error can be arbitrarily larger than the noise level, since $\mathcal{L}(\hat{h}_\beta) = \Omega(1)$ even as $p \to 0$. Note that since the benign overfitting result for $\beta = d$ holds over any distribution and target function (under the same regularity assumptions), the fact that the lower bound of Theorem 5.1 approaches 0 as $\beta \to d$ is to be expected.

**Remark 5.2.** *Interestingly, the only role played by $d$ in the proofs of Theorems 4.1 and 5.1 is the fact that locally, the probability mass scales as $\int_{B(\mathbf{x},r)} \mu \asymp r^d$ (for almost all $\mathbf{x}$ and small $r > 0$). Accordingly, when the data distribution is supported on a lower dimensional manifold of dimension $d_{\mathrm{int}} < d$, the result suggests that tempered overfitting occurs whenever $\beta > d_{\mathrm{int}}$, and that catastrophic overfitting can occur whenever $\beta < d_{\mathrm{int}}$. Although we do not attempt to formalize it in this paper,[3] we conjecture that in general the parameter $d$ can be replaced by $d_{\mathrm{int}}$ in all our results. Since $d_{\mathrm{int}}$ generally can only be estimated, it suggests a potential practical implication: Setting $\beta$ to an over-estimate of $d_{\mathrm{int}}$ is less harmful than under-estimating it, as the former leads to tempered overfitting whereas the latter may lead to catastrophic overfitting. This is further supported by our experiments in Section 6.*

We provide below a sketch of the main ideas of the proof, which is provided in Appendix C. Notably, the main idea behind the proof is quite different from that of Theorem 4.1. There, the analysis was highly *local*, i.e. for every test point $\mathbf{x}$ we showed that we can restrict our analysis to a small neighborhood around that point. In contrast, the reason we will obtain catastrophic overfitting for $\beta < d$ is precisely that the predictor is too *global*, as we will see that for every test point $\mathbf{x}$, all points $\mathbf{x}_i$ in the training set have a non-negligible effect on $\hat{h}_\beta(\mathbf{x})$. Our proof essentially shows that whenever a small region of constant probability mass is surrounded by the opposite label, the predictor will mislabel it, incurring a constant error. Our construction is therefore quite generic, and we expect the same intuition to extend to many target functions $f^*$. The full proof can be found in the appendix.

*Proof sketch of Theorem 5.1.* We will construct an explicit distribution and target function for which $\hat{h}_\beta$ exhibits catastrophic overfitting. The distribution we consider consists of an inner ball of constant probability mass labeled $-1$, and an outer annulus labeled $+1$, as illustrated in Figure 2. Specifically, we denote $c := c(\beta, d) = C_1^{\beta} \cdot (1 - \beta/d)$ for some absolute constant $C_1 > 0$ to be specified later, and consider the following density and target function:

$$\mu_c(\mathbf{x}) = \begin{cases} \frac{c}{\mathrm{Vol}\left(B\left(\mathbf{0}, \frac{1}{4}\right)\right)} & \text{if } \|\mathbf{x}\| \leq \frac{1}{4} \\ \frac{1-c}{\mathrm{Vol}\left(B(\mathbf{0},1) \setminus B\left(\mathbf{0}, \frac{3}{4}\right)\right)} & \text{if } \frac{3}{4} \leq \|\mathbf{x}\| \leq 1 \ , \\ 0 & \text{else} \end{cases} \qquad f^*(\mathbf{x}) = \begin{cases} -1 & \text{if } \|\mathbf{x}\| \leq \frac{1}{4} \\ 1 & \text{else} \end{cases} \ .$$

We consider a test point $\mathbf{x}$ with $\|\mathbf{x}\| \leq \frac{1}{4}$, and will show that for sufficiently large $m$, with high probability $\mathbf{x}$ will be misclassified as $+1$. This implies the desired result, since then

$$\mathcal{L}(\hat{h}_\beta) \gtrsim \Pr_{\mathbf{x}} \left[ \|\mathbf{x}\| \leq \frac{1}{4} \right] = c \ .$$

To that end, we decompose

$$\sum_{i=1}^{m} \frac{y_i}{\|\mathbf{x} - \mathbf{x}_i\|^{\beta}} = \sum_{i:\|\mathbf{x}_i\| \leq \frac{1}{4}} \frac{y_i}{\|\mathbf{x} - \mathbf{x}_i\|^{\beta}} + \sum_{i:\|\mathbf{x}_i\| \geq \frac{3}{4}} \frac{y_i}{\|\mathbf{x} - \mathbf{x}_i\|^{\beta}}$$

$$\geq \underbrace{- \sum_{i:\|\mathbf{x}_i\| \leq \frac{1}{4}} \frac{1}{\|\mathbf{x} - \mathbf{x}_i\|^{\beta}}}_{=:T_1} + \underbrace{\sum_{i:\|\mathbf{x}_i\| \geq \frac{3}{4}} \frac{1 - 2p}{\|\mathbf{x} - \mathbf{x}_i\|^{\beta}}}_{=:T_2} + \underbrace{\sum_{i:\|\mathbf{x}_i\| \geq \frac{3}{4}} \frac{y_i - 1 + 2p}{\|\mathbf{x} - \mathbf{x}_i\|^{\beta}}}_{=:T_3}, \quad (3)$$

---

[3]This should not be difficult in principle, but the proofs would become substantially more technical when the manifold is non-linear.

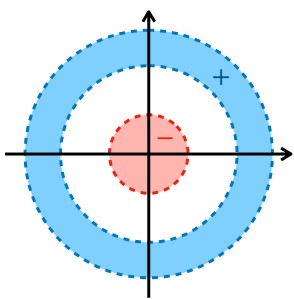

Figure 2: Illustration of the lower bound construction used in the proof of Theorem 5.1. When $\beta < d$, the inner circle will be misclassified as $+1$ with high probability, inducing constant error.

where $T_1$ crudely bounds the contribution of points in the inner circle, $T_2$ is the expected contribution of outer points labeled 1, and $T_3$ is a perturbation term. Noting that $T_2 > 0$, our goal is to show that $T_2$ dominates the expression above, implying that Eq. (3) is positive and thus $h_\beta(\mathbf{x}) = 1$.

Let $k := \left|\{i : \|\mathbf{x}_i\| \leq \frac{1}{4}\}\right|$ denote the number of points inside the inner ball, and note that we can expect $k \approx \mathbb{E}[k] = cm$. To bound $T_1$, we express its distribution using exponential random variables in a manner that is similar to the proof of Theorem 4.1. Specifically, for standard exponential random variables $E_1, \ldots, E_m \overset{i.i.d.}{\sim} \exp(1)$, we show that with high probability

$$-T_1 \gtrsim -\sum_{i:\|\mathbf{x}_i\| \leq \frac{1}{4}} \frac{\left(\sum_{i=1}^m E_i\right)^{\beta/d}}{\left(\frac{1}{4c^{1/d}}\right)^\beta \left(\sum_{j=1}^i E_j\right)^{\beta/d}} \gtrsim_{(1)} -c^{\beta/d} 4^\beta \cdot m^{\beta/d} \cdot \sum_{i=k+1}^m \frac{1}{i^{\beta/d}}$$

$$\gtrsim -c^{\beta/d} 4^\beta \cdot m^{\beta/d} \cdot \frac{k^{1-\beta/d}}{1-\beta/d} \gtrsim_{(2)} -cm \cdot \frac{4^\beta}{(1-\beta/d)} \,,$$

where (1) uses concentration bounds on the sums of exponential random variables to argue that $\sum_{j=1}^i E_j \approx i$, and (2) follows from showing $k \approx cm$.

To show that $T_2$ is sufficiently large, we use the fact that $\|\mathbf{x} - \mathbf{x}_i\| \leq \|\mathbf{x}\| + \|\mathbf{x}_i\| \leq \frac{5}{4}$, and that $\left|\{i : \|\mathbf{x}_i\| \geq \frac{3}{4}\}\right| \approx (1-c)m \geq \frac{1}{2}m$ with high probability to obtain

$$T_2 = \sum_{i:\|\mathbf{x}_i\| \geq \frac{3}{4}} \frac{1-2p}{\|\mathbf{x} - \mathbf{x}_i\|^\beta} \geq (1-2p)\left|\left\{\mathbf{x}_i : \|\mathbf{x}_i\| \geq \frac{3}{4}\right\}\right| \cdot \left(\frac{4}{5}\right)^\beta \gtrsim m \cdot \left(\frac{4}{5}\right)^\beta.$$

Lastly, we show that $T_3$ is asymptotically negligible, by noting that $\mathbb{E}[T_3] = 0$ hence $T_3 = o(m)$ with high probability by Hoeffding's inequality. Thus Eq. (3) becomes

$$\hat{h}_\beta(\mathbf{x}) = \text{sign}\left[\sum_{i=1}^m \frac{y_i}{\|\mathbf{x} - \mathbf{x}_i\|^\beta}\right] \gtrsim \text{sign}\left[m\left(\left(\frac{4}{5}\right)^\beta - \frac{c \cdot 4^\beta}{1-\beta/d}\right)\right].$$

Overall, we see that the right-hand side above is positive as long as $c = C_1^\beta \cdot \left(1 - \frac{\beta}{d}\right) < \frac{1-\beta/d}{5^\beta}$, or equivalently $C_1 < \frac{1}{5}$, meaning that $\hat{h}_\beta(\mathbf{x}) = 1$ even though $f^*(\mathbf{x}) = -1$.

$\square$

## 6 Experiments

In this section, we provide numerical simulations that illustrate and complement our theoretical findings. In all experiments, we sample $m$ datapoints according to some distribution, flip each label independently with probability $p$, and plot the clean test error of $\hat{h}_\beta$ for various values of $\beta$. We ran each experiment 50 times, and plotted the average error surrounded by a 95% confidence interval.

### 6.1 Synthetic data

We start by discussing several experiments with synthetic data distributions.

**Warm up: one dimensional data.** In our first experiment, we considered data in dimension $d = 1$ distributed according to the construction considered in the proof of Theorem 5.1. In particular, we consider

$$\mathcal{D}_x = \frac{1}{10} \cdot \text{Unif}\left((0, \tfrac{1}{4})\right) + \frac{9}{10} \cdot \text{Unif}\left((\tfrac{3}{4}, 1)\right), \qquad f^*(x) = \begin{cases} -1 & \text{if } x \in (0, \tfrac{1}{4}) \\ 1 & \text{else} \end{cases}. \tag{4}$$

In Figure 3, on the left we plot the results for $m = 2000$ and various values of $p$, and on the right we fix $p = 0.04$ and vary $m$.

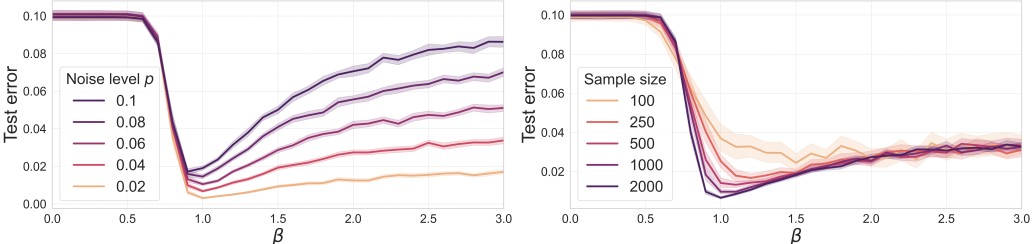

Figure 3: The classification error of $\hat{h}_\beta$ for varying values of $\beta$, with data in dimension $d = 1$ given by Eq. (4). On the left, $m = 2000$ is fixed, $p$ varies. On the right, $p = 0.04$ is fixed, $m$ varies. Best viewed in color.

As seen in Figure 3, the generalization is highly asymmetric with respect to $\beta$. For $\beta < 1$, the test error degrades independently of the noise level $p$, and quickly reaches $0.1$ in all cases, illustrating that the predictor errors on the negative labels (which have $0.1$ probability mass). On the other hand, for $\beta > 1$, the test error exhibits a gradual deterioration. Moreover, we see this deterioration is controlled by the noise level $p$, matching our theoretical finding. The right figure illustrates all of the discussed phenomena hold similarly for moderate sample sizes, which complements our asymptotic analysis.

**Spherical data.** In our second experiment, we consider a similar distribution over the unit sphere $\mathbb{S}^2 \subset \mathbb{R}^3$, where the inner negatively labeled region is a spherical cap. In particular, consider the spherical cap defined by $A := \left\{ \mathbf{x} = (x_1, x_2, x_3) \in \mathbb{S}^2 \mid x_3 > \sqrt{3}/2 \right\}$, and let

$$\mathcal{D}_\mathbf{x} = \frac{1}{10} \cdot \text{Unif}(A) + \frac{9}{10} \cdot \text{Unif}(\mathbb{S}^2 \setminus A), \qquad f^*(\mathbf{x}) = \begin{cases} -1 & \text{if } \mathbf{x} \in A \\ 1 & \text{else} \end{cases}. \tag{5}$$

In Figure 4, on the left we plot the results for $m = 2000$ and various values of $p$, and on the right we fix $p = 0.04$ and vary $m$.

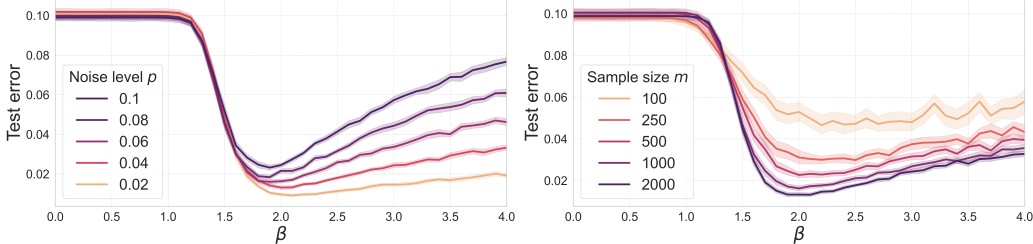

Figure 4: The classification error of $\hat{h}_\beta$ for varying values of $\beta$, with data on $\mathbb{S}^2 \subset \mathbb{R}^3$ given by Eq. (5). On the left, $m = 2000$ is fixed, $p$ varies. On the right, $p = 0.04$ is fixed, $m$ varies. Best viewed in color.

As seen in Figure 4, the same asymmetric phenomenon holds in which overly large $\beta$ are more forgiving than overly small $\beta$, especially in low noise regimes. The main difference between the first

and second experiment is that the optimal "benign" exponent in the second case is $\beta = 2$, matching the *intrinsic* dimension of the sphere, even though the data is embedded in 3-dimensional space. This agrees with our conjecture that for distributions with low intrinsic dimension $d_{\text{int}} < d$, the overfitting behavior depends on $d_{\text{int}}$ rather than $d$ (as discussed in Remark 5.2).

In Appendix E we provide an extension of the spherical data experiment, in which the inputs are corrupted by Gaussian noise. As the noise variance increases, hence the dataset is drawn away from having a low intrinsic dimension, the $\beta$ value with minimal test error gradually increases from 2 to 3. This illustrates a robustness to input-noise which is prevalent in practice, complementing an aspect that our current formal results do not cover.

## 6.2  Intrinsic Dimension of MNIST

Next, we consider an experiment in which the data consists of images of handwritten $0$ and $1$ digits from the MNIST dataset. In Figure 5, on the left we plot the results with respect to the entire training set $m = 12,665$ and various values of $p$, and on the right we fix $p = 0.1$ and vary $m$.

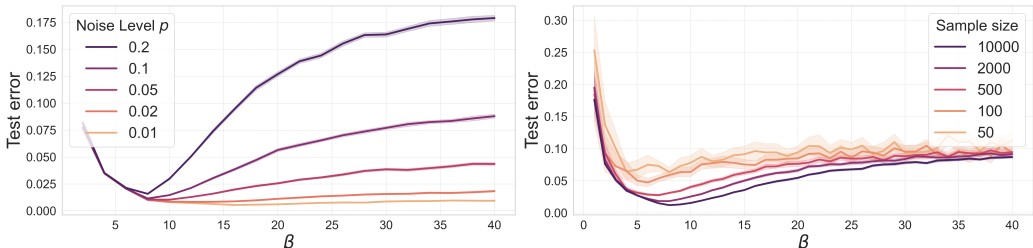

Figure 5: The classification error of $\hat{h}_\beta$ for varying values of $\beta$, with respect to MNIST's $0/1$ data. On the left, $m$ is fixed to the entire train set, $p$ varies. On the right, $p = 0.1$ is fixed, $m$ varies. Best viewed in color.

As seen in Figure 5, the same asymmetric phenomenon demonstrated by both our theory as well as the synthetic experiments, clearly holds once more. It is interesting to note that each image in MNIST is of size $28 \times 28 = 784$ pixels, and so the extrinsic dimension is $784$. Nonetheless, the optimal exponent is roughly $\beta \approx 8 \ll 784$, which matches an estimate of the intrinsic dimension of MNIST measured by Pope et al. [45]. Moreover, as seen on the right, the asymptotic phenomenon manifests quite clearly already for small samples sizes, and only becomes more pronounced as the number of samples increases.

## 7  Discussion

In this work, we characterized the generalization behavior of the NW interpolator for any choice of the hyperparameter $\beta$. Specifically, NW interpolates in a tempered manner when $\beta > d$, exhibits benign overfitting when $\beta = d$, and overfits catastrophically when $\beta < d$. This substantially extends the classical analysis of this method, which only focused on consistency. In addition, it indicates that the NW interpolator is much more tolerant to over-estimating $\beta$ as opposed to under-estimating it.

Our analysis and experiments both suggest that the dependence on $d$ arises from the assumption that the distributions considered here have a density in $\mathbb{R}^d$, and that more generally over-estimating the intrinsic dimension of the data is preferable to under-estimating it when setting $\beta$.

Overall, our results highlight how intricate generalization behaviors, including the full range from benign through tempered to catastrophic overfitting, can already appear in simple and well-known interpolating learning rules. We hope these results will further motivate revisiting other fundamental learning rules using this modern viewpoint, going beyond the classical consistency-vs.-inconsistency dichotomy.

## Acknowledgments and Disclosure of Funding

This research is supported in part by European Research Council (ERC) grant 754705, by the Israeli Council for Higher Education (CHE) via the Weizmann Data Science Research Center and by research grants from the Estate of Harry Schutzman and the Anita James Rosen Foundation. GK is supported by an Azrieli Foundation graduate fellowship.

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

## A  Notation and order statistics

We start by introducing some notation that we will use throughout the proofs to follow. We denote $X_1 \asymp X_2$ to abbreviate $X_1 = \Theta(X_2)$, $X_1 \lesssim X_2$ to abbreviate $X_1 = \mathcal{O}(X_2)$ and $X_1 \gtrsim X_2$ to abbreviate $X_1 = \Omega(X_2)$. Throughout the proofs we let $\alpha := \beta/d$, and abbreviate $h = \hat{h}_\beta$. Given some $\mathbf{x} \in \mathrm{supp}(\mu)$ and $m \in \mathbb{N}$, we consider the one-dimensional random variables

$$W_i^{\mathbf{x}} := V_d^\alpha \, \|\mathbf{x} - \mathbf{x}_i\|^\beta \ ,$$

where $V_d$ is the volume of the $d$ dimensional unit ball, and the randomness is over $\mathbf{x}_i$. We let $F_{\mathbf{x}}$ be the CDF of $W_i^{\mathbf{x}}$ (which is clearly the same for all $i \in [m]$). We also let $U_i \sim U([0,1])$, $i \in [m]$ be standard uniform random variables, and denote by $W_{(i)}^{\mathbf{x}}$ and $U_{(i)}$ the ordered versions of the $W_i^{\mathbf{x}}$s and $U_i$s respectively, namely

$$W_{(1)}^{\mathbf{x}} \le W_{(2)}^{\mathbf{x}} \le \cdots \le W_{(m)}^{\mathbf{x}} \ ,$$
$$U_{(1)} \le U_{(2)} \le \cdots \le U_{(m)} \ .$$

We will often omit the superscript/subscript $\mathbf{x}$ where it is clear by context and use the notations $W_i, W_{(i)}$ and $F$ to denote $W_i^{\mathbf{x}}, W_{(i)}^{\mathbf{x}}$ and $F_{\mathbf{x}}$ respectively. Lastly, we let $F^{-1} : [0,1] \to [0,1]$, $F^{-1}(t) = \inf\{s : F(s) \ge t\}$ be the quantile function, and note that it satisfies $F(w) \le u$ if and only if $w \le F^{-1}(u)$.

**Lemma A.1** (46, Theorem 2.1). *For any $i \in [m]$, $\mathbf{x} \in \mathbb{R}^d$ : $W_i^{\mathbf{x}} \stackrel{d}{=} F_{\mathbf{x}}^{-1}(U_i)$.*

Note that since $(W_i)_{i\in[m]}$ are independent, the lemma above further applies to the joint distribution, and to the joint distribution of the order statistics (see e.g. 46, Example 2.3). Since we will use this often, we state this as a separate lemma.

**Lemma A.2.** *For any $\mathbf{x} \in \mathbb{R}^d$ : $(W_{(i)}^{\mathbf{x}})_{i\in[m]} \stackrel{d}{=} \left(F_{\mathbf{x}}^{-1}(U_{(i)})\right)_{i\in[m]}$ jointly.*

The behavior of $U_{(i)}$ is best understood through the following lemma.

**Lemma A.3** (47, Chapter 8, Proposition 1). *Let $E_1, \dots E_{m+1}$ be i.i.d. standard exponential random variables. Then*

$$\left(U_{(1)}, \dots, U_{(m)}\right) \sim \frac{1}{\sum_{i=1}^{m+1} E_i} \left(\sum_{i=1}^{1} E_i, \dots, \sum_{i=1}^{m} E_i\right) \ .$$

## B  Proof of Theorem 4.1

Throughout the proof, we will use the notation introduced in Appendix A.

**Lemma B.1.** *For almost every $\mathbf{x} \in \mathbb{R}^d$ and $\epsilon > 0$, there exists $\delta_{\mathbf{x}} > 0$ such that for any $u \le V_d^\alpha \delta_{\mathbf{x}}^\beta$ :*

$$\frac{2}{3\mu(\mathbf{x})} u^\alpha \le F^{-1}(u) \le \frac{2}{\mu(\mathbf{x})} u^\alpha \ .$$

*Proof.* By the Lebesgue differentiation theorem (cf. 48, Chapter 3), for almost every $\mathbf{x} \in \mathbb{R}^d$ and $\epsilon > 0$, there exists some $\delta_{\mathbf{x}} > 0$ such that

$$\sup_{0 < r \le \delta_{\mathbf{x}}} \left| \frac{\int_{B(\mathbf{x},r)} \mu(\mathbf{z})d\mathbf{z}}{V_d r^d} - \mu(\mathbf{x}) \right| \le \epsilon\mu(\mathbf{x}).$$

In particular, for any $0 < u \le V_d^\alpha \delta_{\mathbf{x}}^\beta$, taking $r = \frac{u^{1/\beta}}{V_d^{1/d}}$ (which in particular satisfies $r \le \delta_{\mathbf{x}}$) we have

$$\left| \frac{\int_{B(\mathbf{x},r)} \mu(\mathbf{z})d\mathbf{z}}{u^{\frac{1}{\alpha}}} - \mu(\mathbf{x}) \right| \le \epsilon\mu(\mathbf{x}).$$

As a result,

$$F(w) = \Pr_{\mathbf{z}}\left(V_d^\alpha \|\mathbf{x}-\mathbf{z}\|^\beta \le w\right) = \Pr_{\mathbf{z}}\left(\|\mathbf{x}-\mathbf{z}\| \le \frac{w^{1/\beta}}{V_d^{1/d}}\right)$$

$$= \int_{B(\mathbf{x},r)} \mu(\mathbf{z})d\mathbf{z} \in \left[(1-\epsilon)\mu(\mathbf{x})w^{1/\alpha}, (1+\epsilon)\mu(\mathbf{x})w^{1/\alpha}\right].$$

The result readily follows by plugging $\epsilon = \frac{1}{2}$ and inverting. $\qquad\square$

**Lemma B.2.** *For any $k \in \mathbb{N}$, $\alpha > 1$ and almost every $\mathbf{x} \in \mathbb{R}^d$ there exists a constant $\tilde{C}(\mathbf{x}, k, \alpha)$ such that as long as $m \ge \tilde{C}(\mathbf{x}, k, \alpha)$, the following holds: If the $k$ nearest neighbors of $\mathbf{x}$ are all labeled the same $y_{(1)} = \cdots = y_{(k)}$, then $h(\mathbf{x}) = y_{(1)}$ with probability at least $1 - c_1 \exp(-c_2 k) - \exp(-c_\alpha k^{1-\frac{1}{\alpha}})$ over the randomness of $(\mathbf{x}_i)_{i=1}^m$.*

*Proof.* Given $\mathbf{x}$, let $\delta = \delta_{\mathbf{x}} > 0$ be the radius given by Lemma B.1, and assume without loss of generality that $\delta$ is sufficiently small so that $f^*$ is constant over $B(\mathbf{x}, \delta)$ (otherwise replace it by the smaller radius given by Assumption 3.1). Note that for all indices $i$ such that $W_{(i)} \le \delta$, it holds that $y_i$ are independent variables (that equal $f^*(\mathbf{x})$ with probability $1-p$). Furthermore, given $k \in \mathbb{N}$, we assume $m$ is sufficiently large so that the $k$ nearest neighbors of $\mathbf{x}$ all lie in $B(\mathbf{x}, \delta)$ with probability at least $1 - \exp(-k)$. Under this likely event, we decompose

$$\sum_{i=1}^m \frac{y_i}{W_i} = \sum_{i:W_{(i)} \le \delta} \frac{y_i}{W_i} + \sum_{i:W_{(i)} > \delta} \frac{y_i}{W_i} \overset{d}{=} \underbrace{\sum_{i=1}^k \frac{y_{(i)}}{F^{-1}(U_{(i)})}}_{(I)} + \underbrace{\sum_{i=k+1}^{|S_{\mathbf{x}} \cap B(\mathbf{x},\delta)|} \frac{y_{(i)}}{F^{-1}(U_{(i)})}}_{(II)} + \underbrace{\sum_{i:W_{(i)} > \delta} \frac{y_i}{W_i}}_{(III)}.$$

We will show that whenever $y_{(1)} = \cdots = y_{(k)}$, then with high probability $(I)$ is the dominant term in the sum above. We start by noting that if $y_{(1)} = \cdots = y_{(k)}$, then $(I) = y_{(1)} \sum_{i=1}^k \frac{1}{F^{-1}(U_{(i)})}$, thus

$$|(I)| = \sum_{i=1}^k \frac{1}{F^{-1}(U_{(i)})} \ge \frac{1}{F^{-1}(U_{(1)})} \ge \frac{\mu(\mathbf{x})}{2U_{(1)}^\alpha} \overset{d}{=} \frac{\mu(\mathbf{x})}{2}\left(\sum_{i=1}^{m+1} E_i\right)^\alpha \cdot \frac{1}{E_1^\alpha}. \qquad (6)$$

Similarly,

$$|(II)| \le 2\mu(\mathbf{x}) \sum_{i=k+1}^{|S_{\mathbf{x}} \cap B(\mathbf{x},\delta)|} \frac{1}{U_{(i)}^\alpha} \le 2\mu(\mathbf{x}) \sum_{i=k+1}^m \frac{1}{U_{(i)}^\alpha} \overset{d}{=} 2\mu(\mathbf{x})\left(\sum_{i=1}^{m+1} E_i\right)^\alpha \cdot \sum_{i=k+1}^m \frac{1}{(\sum_{j=1}^i E_j)^\alpha}$$

So the probability of $|(II)| < \frac{1}{2}|(I)|$ is at least the probability of the event in which $8\sum_{i=k+1}^m \frac{1}{(\sum_{j=1}^i E_j)^\alpha} < \frac{1}{E_1^\alpha}$. To see this event is indeed likely, we apply Lemma D.1 to get that with probability at least $1 - c_1 \exp(-c_2 k)$, for all $i \ge k+1$ : $\frac{1}{(\sum_{j=1}^i E_j)^\alpha} \le \frac{1}{(i/2)^\alpha}$, and therefore under this event we get

$$8\sum_{i=k+1}^m \frac{1}{(\sum_{j=1}^i E_j)^\alpha} \le 8 \cdot 2^\alpha \sum_{i=k+1}^\infty \frac{1}{i^\alpha} \le 8 \cdot 2^\alpha \int_k^\infty t^{-\alpha}dt = \frac{8 \cdot 2^\alpha}{(\alpha-1)k^{\alpha-1}}.$$

The latter is smaller than $1/E_1^\alpha$ as long as $E_1 \le \frac{(\alpha-1)^{1/\alpha}k^{1-1/\alpha}}{2 \cdot 8^{1/\alpha}}$, which by definition, occurs with probability $1 - \exp\left[-\frac{(\alpha-1)^{1/\alpha}}{2 \cdot 8^{1/\alpha}}k^{1-1/\alpha}\right]$. To complete the proof, we note that $|(III)| \le \frac{m}{\delta}$ is asymptotically negligible for sufficiently large $m$, since using Eq. (6) we see that $|I| \ge \frac{\mu(\mathbf{x})}{2E_1^\alpha}\left(\sum_{i=1}^{m+1} E_i\right)^\alpha \ge \frac{\mu(\mathbf{x})}{2E_1^\alpha}(m/2)^\alpha = \omega(m)$ with probability at least $1 - c_1 \exp(-c_2 m)$ by Lemma D.1.

$\qquad\square$

**Lemma B.3.** *Given $\mathbf{x} \in \mathbb{R}^d$, let $A_{\mathbf{x}}^k$ be the event in which all of $\mathbf{x}$'s $k$ nearest neighbors $(\mathbf{x}_{(i)})_{i=1}^k$ satisfy $f^*(\mathbf{x}_i) = f^*(\mathbf{x})$. Then for any fixed $k$, it holds for almost every $\mathbf{x} \in \text{supp}(\mu)$ that $\lim_{m\to\infty} \Pr[A_{\mathbf{x}}^k] = 1$.*

*Proof.* Let $\mathbf{x} \in \mathrm{supp}(\mu)$ be such that $\mu$ is continuous at $\mathbf{x}$ (which holds for a full measure set by assumption). Since $\mu(\mathbf{x}) > 0$, then there exists $\rho > 0$ so that $\mu|_{B(\mathbf{x},\rho)} > 0$, and assume $\rho$ is sufficiently small so that $f^*|_{B(\mathbf{x},\rho)} = f^*(\mathbf{x})$. Note that $B(\mathbf{x}, \rho)$ has some positive probability mass which we denote by $\phi := \int_{B(\mathbf{x},\rho)} \mu$. Under this notation, we see that

$$\Pr[\neg A_{\mathbf{x}}^k] \le \Pr\left[|\{\mathbf{x}_1, \ldots, \mathbf{x}_m\} \cap B(\mathbf{x}, \rho)| < k\right] = \Pr[\mathrm{Binomial}(m, \phi) < k] \xrightarrow{m \to \infty} 0 .$$

$\square$

**Proof of Theorem 4.1** We start by proving the upper bound. Let $k := \log^{\frac{\alpha}{\alpha-1}}(1/p)$, and for any $\mathbf{x} \in \mathbb{R}^d$, consider the event $A_{\mathbf{x}}^k$ in which $\mathbf{x}$'s $k$ nearest neighbors $(\mathbf{x}_{(i)})_{i=1}^k$ satisfy $f^*(\mathbf{x}_i) = f^*(\mathbf{x})$ (as described in Lemma B.3). Using the law of total expectation, we have that

$$\begin{aligned}
\mathbb{E}_S\left[\Pr_{\mathbf{x}}(h(\mathbf{x}) \ne f^*(\mathbf{x}))\right] &= \mathbb{E}_S \mathbb{E}_{\mathbf{x}}[\mathbb{1}\{h(\mathbf{x}) \ne f^*(\mathbf{x})\}] \\
&= \mathbb{E}_{\mathbf{x}} \mathbb{E}_S\left[\mathbb{1}\{h(\mathbf{x}) \ne f^*(\mathbf{x})\}\right] \\
&= \mathbb{E}_{\mathbf{x}}\left[\mathbb{E}_S\left[\mathbb{1}\{h(\mathbf{x}) \ne f^*(\mathbf{x})\} \mid A_{\mathbf{x}}^k\right] \cdot \Pr_S[A_{\mathbf{x}}^k]\right] \\
&\quad + \mathbb{E}_{\mathbf{x}}\left[\mathbb{E}_S\left[\mathbb{1}\{h(\mathbf{x}) \ne f^*(\mathbf{x})\} \mid \neg A_{\mathbf{x}}^k\right] \Pr_S[\neg A_{\mathbf{x}}^k]\right] \qquad (7) \\
&\le \mathbb{E}_{\mathbf{x}} \mathbb{E}_S\left[\mathbb{1}\{h(\mathbf{x}) \ne f^*(\mathbf{x})\} \mid A_{\mathbf{x}}^k\right] + \mathbb{E}_{\mathbf{x}} \Pr_S[\neg A_{\mathbf{x}}^k] .
\end{aligned}$$

Note that by Lemma B.3 $\lim_{m \to \infty} \Pr_S[\neg A_{\mathbf{x}}^k] = 0$, and therefore it remains to bound the first summand above.

To that end, we continue by temporarily fixing $\mathbf{x}$. Denote by $B_{\mathbf{x}}^k$ the event in which $\mathbf{x}$'s $k$ nearest neighbors are all labeled correctly (namely, their labels were not flipped), and note that $\Pr_S[B_{\mathbf{x}}^k] = (1-p)^k \ge 1 - kp$, hence $\Pr_S[\neg B_{\mathbf{x}}^k] < kp$. By Lemma B.2 we also know that for sufficiently large $m$:

$$\Pr_S[h(\mathbf{x}) \ne f^*(\mathbf{x}) \mid A_{\mathbf{x}}^k, B_{\mathbf{x}}^k] \le c_1 \exp(-c_2 k) + \exp(-c_\alpha k^{1-\frac{1}{\alpha}}) .$$

Therefore,

$$\begin{aligned}
\mathbb{E}_S[\mathbb{1}\{h(\mathbf{x}) \ne f^*(\mathbf{x})\} \mid A_{\mathbf{x}}^k] &= \Pr_S[h(\mathbf{x}) \ne f^*(\mathbf{x}) \mid A_{\mathbf{x}}^k] \\
&= \Pr_S[h(\mathbf{x}) \ne f^*(\mathbf{x}) \mid A_{\mathbf{x}}^k, B_{\mathbf{x}}^k] \cdot \Pr_S[B_{\mathbf{x}}^k] \\
&\quad + \Pr_S[h(\mathbf{x}) \ne f^*(\mathbf{x}) \mid A_{\mathbf{x}}^k, \neg B_{\mathbf{x}}^k] \cdot \Pr_S[\neg B_{\mathbf{x}}^k] \\
&\le \left(c_1 \exp(-c_2 k) + \exp(-c_\alpha k^{1-\frac{1}{\alpha}})\right) \cdot 1 + 1 \cdot kp \\
&\le C_\alpha \log^{\frac{\alpha}{\alpha-1}}(1/p) \log(p) ,
\end{aligned}$$

where the last inequality follows by our assignment of $k$. Since this is true for any $\mathbf{x}$, it is also true in expectation over $\mathbf{x}$, thus completing the proof of the upper bound.

We proceed to prove the lower bound. We consider $A_{\mathbf{x}}^k$ to be the same event as before, yet now we set $k := k_\alpha = \left(\frac{8 \cdot 2^\alpha}{\alpha-1}\right)^{\frac{1}{\alpha-1}}$. By lower bounding Eq. (7) (instead of upper bounding it as before), we obtain

$$\mathbb{E}_S\left[\Pr_{\mathbf{x}}(h(\mathbf{x}) \ne f^*(\mathbf{x}))\right] \ge \mathbb{E}_{\mathbf{x}}[\underbrace{\mathbb{E}_S[\mathbb{1}\{h(\mathbf{x}) \ne f^*(\mathbf{x})\} \mid A_{\mathbf{x}}^k]}_{(\star)} \cdot \Pr_S[A_{\mathbf{x}}^k]] - \mathbb{E}_{\mathbf{x}} \Pr_S[\neg A_{\mathbf{x}}^k] .$$

As $\lim_{m \to \infty} \Pr_S[A_{\mathbf{x}}^k] = 1$ and $\lim_{m \to \infty} \Pr_S[\neg A_{\mathbf{x}}^k] = 0$ by Lemma B.3, it once again remains to bound $(\star)$.

To that end, we temporarily fix $\mathbf{x}$, denote by $D_{\mathbf{x}}^k$ the event in which the labels of $\mathbf{x}$'s $k$ nearest neighbors were are all flipped. Note that since the label flips are independent of the location of the datapoints, it holds that $\Pr_S[D_{\mathbf{x}}^k \mid A_{\mathbf{x}}^k] = \Pr_S[D_{\mathbf{x}}^k] = p^k$. By Lemma B.2 we also know that for sufficiently large $m$:

$$\Pr_S[h(\mathbf{x}) \ne f^*(\mathbf{x}) \mid A_{\mathbf{x}}^k, D_{\mathbf{x}}^k] \ge 1 - c_1 \exp(-c_2 k) - \exp(-c_\alpha k^{1-\frac{1}{\alpha}}) .$$

Therefore,

$$
\begin{aligned}
\mathbb{E}_S[\mathbb{1}\left\{h(\mathbf{x}) \neq f^*(\mathbf{x})\right\} \mid A_{\mathbf{x}}^k] &= \Pr_S[h(\mathbf{x}) \neq f^*(\mathbf{x}) \mid A_{\mathbf{x}}^k] \\
&\geq \Pr_S[h(\mathbf{x}) \neq f^*(\mathbf{x}) \ \wedge \ D_{\mathbf{x}}^k \mid A_{\mathbf{x}}^k] \\
&= \Pr_S[h(\mathbf{x}) \neq f^*(\mathbf{x}) \mid A_{\mathbf{x}}^k, D_{\mathbf{x}}^k] \cdot \Pr[D_{\mathbf{x}}^k \mid A_{\mathbf{x}}^k] \\
&\geq \left(1 - c_1 \exp(-c_2 k) - \exp(-c_\alpha k^{1-\frac{1}{\alpha}})\right) p^k \\
&\geq c_\alpha p^k \, ,
\end{aligned}
$$

is due to our assignment of $k$ (and the explicit form of $c_\alpha$ in Lemma B.2).

## C   Proof of Theorem 5.1

**Setting for the proof.**   Throughout the proof, we will use the notation introduced in Appendix A. We start by specifying the target function and distribution for which we will prove that catastrophic overfitting occurs. We will consider a slightly more general version than mentioned in the main text. Fix $R, r, c > 0$ that satisfy $R > 3r$. We define a distribution on $B(\mathbf{0}, R)$ whose density is given by

$$
\mu(\mathbf{x}) := \begin{cases} \frac{c}{\mathrm{Vol}(B(\mathbf{0},r))} & \|\mathbf{x}\| < r \\ \frac{1-c}{\mathrm{Vol}(B(\mathbf{0},R)\backslash B(\mathbf{0},3r))} & 3r \leq \|\mathbf{x}\| \leq R \\ 0 & \text{else} \end{cases} = \begin{cases} \frac{c}{V_d r^d} & \|\mathbf{x}\| < r \\ \frac{1-c}{V_d \cdot (R^d - (3r)^d)} & 3r \leq \|\mathbf{x}\| \leq R \, , \\ 0 & \text{else} \end{cases}
$$

where $V_d$ is the volume of the $d$-dimensional unit ball. We also define the target function

$$
f^*(\mathbf{x}) := \begin{cases} -1 & \|\mathbf{x}\| \leq r \\ 1 & \text{else} \end{cases} .
$$

The main lemma from we derive the proof of Theorem 5.1 is the following:

**Lemma C.1.** *Under setting C suppose that $c$ satisfies*

$$
c \leq \frac{1 - \beta/d}{2400 \left(1 + \frac{R}{r}\right)^\beta}.
$$

*Then there exists some $m_0 \in \mathbb{N}$, such that for any $\mathbf{x} \in B(\mathbf{0}, r)$, $m > m_0$ and $p \in (0, 0.49)$, it holds with probability at least $1 - \tilde{\mathcal{O}}_m \left(\frac{1}{m} + \frac{1}{m^{\frac{1-\beta/d}{\beta/d}}}\right)$ over the randomness of the training set $S$ that*

$$
\hat{h}_\beta(\mathbf{x}) = 1.
$$

We temporarily defer the proof of Lemma C.1, and start by showing that it easily implies the theorem:

*Proof.*   **of Theorem 5.1** Fix $R > 3r$, let $c = \frac{1-\beta/d}{2400(1+\frac{R}{r})^\beta}$ and consider the distribution and target function given by Setting C. Using the law of total expectation, we have that

$$
\begin{aligned}
\mathbb{E}_S\left[\Pr_{\mathbf{x}}(h(\mathbf{x}) \neq f^*(\mathbf{x}))\right] &= \mathbb{E}_S \mathbb{E}_{\mathbf{z}}[\mathbb{1}\left\{h(\mathbf{z}) \neq f^*(\mathbf{z})\right\}] \\
&= \mathbb{E}_{\mathbf{z}} \mathbb{E}_S\left[\mathbb{1}\left\{h(\mathbf{z}) \neq f^*(\mathbf{z})\right\}\right] \\
&\geq \mathbb{E}_{\mathbf{z}}\left[\mathbb{E}_S\left[\mathbb{1}\left\{h(\mathbf{z}) \neq f^*(\mathbf{z})\right\}\right] \mid \mathbf{z} \in B(\mathbf{0}, r)\right] \cdot \Pr(\mathbf{z} \in B(\mathbf{0}, r)) \\
&= \mathbb{E}_{\mathbf{z}}\left[\Pr_S\left(\mathbb{1}\left\{h(\mathbf{z}) \neq f^*(\mathbf{z})\right\} \mid \mathbf{z} \in B(\mathbf{0}, r)\right)\right] \cdot \Pr(\mathbf{z} \in B(\mathbf{0}, r)) \\
&\geq_{(*)} c \left(1 - \tilde{\mathcal{O}}_m \left(\frac{1}{m} + \frac{1}{m^{\frac{1-\beta/d}{\beta/d}}}\right)\right)
\end{aligned}
$$

where $(*)$ follows from Lemma C.1. This completes the proof by sending $m \to \infty$.   $\square$

## C.1 Proof of Lemma C.1

Fix some $\mathbf{x}$ with $\|\mathbf{x}\| < r$, we will show that for sufficiently large $m$, with high probability $\mathbf{x}$ will be misclassified as $+1$. To that end, we decompose

$$\sum_{i=1}^{m} \frac{y_i}{\|\mathbf{x} - \mathbf{x}_i\|^\beta} = \sum_{i:\|\mathbf{x}_i\| \le r} \frac{y_i}{\|\mathbf{x} - \mathbf{x}_i\|^\beta} + \sum_{i:\|\mathbf{x}_i\| \ge 3r} \frac{y_i}{\|\mathbf{x} - \mathbf{x}_i\|^\beta}$$

$$= \sum_{i:\|\mathbf{x}_i\| \le r} \frac{y_i}{\|\mathbf{x} - \mathbf{x}_i\|^\beta} + \sum_{i:\|\mathbf{x}_i\| \ge 3r} \frac{1 - 2p}{\|\mathbf{x} - \mathbf{x}_i\|^\beta} + \sum_{i:\|\mathbf{x}_i\| \ge 3r} \frac{y_i - 1 + 2p}{\|\mathbf{x} - \mathbf{x}_i\|^\beta}$$

$$\ge -\underbrace{\sum_{i:\|\mathbf{x}_i\| \le r} \frac{1}{\|\mathbf{x} - \mathbf{x}_i\|^\beta}}_{=:T_1} + \underbrace{\sum_{i:\|\mathbf{x}_i\| \ge 3r} \frac{1 - 2p}{\|\mathbf{x} - \mathbf{x}_i\|^\beta}}_{=:T_2} - \underbrace{\left| \sum_{i:\|\mathbf{x}_i\| \ge 3r} \frac{y_i - 1 + 2p}{\|\mathbf{x} - \mathbf{x}_i\|^\beta} \right|}_{=:T_3}, \quad (8)$$

where $T_1$ crudely bounds the contribution of points in the inner circle, $T_2$ is the expected contribution of outer points labeled 1, and $T_3$ is a perturbation term. Let $k_m := |\{i \in [m] \mid \|\mathbf{x}_i\| \le r\}|$ denote the number of training points inside the inner ball. By Lemma C.3, whenever $c \le \frac{1}{2}$ (we will ensure this happens) it holds with probability at least $1 - 2\exp\left(-\frac{m}{8}\right)$ that

$$\frac{cm}{2} \le k_m \le \frac{3cm}{2} \le \frac{3m}{4}. \quad (9)$$

Throughout the rest of the proof, we assume this event indeed occurs.

*Bounding $T_1$:* Using that $\|\mathbf{x}\| < r$ and that the pdf $\mu$ is such that for all $\mathbf{x}_i \notin B(\mathbf{0}, r)$, $\|\mathbf{x} - \mathbf{x}_i\| > 3r - r > 2r$, we have that the $k_m$ nearest neighbors $\mathbf{x}_{(1)}, \ldots, \mathbf{x}_{(k_m)}$ are precisely the points with $\|\mathbf{x}_i\| \le r$.

For any $w \le (2r)^\beta$ and any $\mathbf{z} \in B\left(\mathbf{x}, w^{\frac{1}{\beta}}\right)$ it holds that $\|\mathbf{z}\| \le 3r$, and $\mu(\mathbf{x}) \le \frac{c}{V_d r^d}$. Thus, for such a $w$,

$$F(w) := \Pr_{\mathbf{z}}\left(\|\mathbf{z} - \mathbf{x}\|^\beta \le w\right) = \Pr_{\mathbf{z}}\left(\|\mathbf{z} - \mathbf{x}\| \le w^{\frac{1}{\alpha d}}\right) = \int_{B\left(\mathbf{x}, w^{\frac{1}{\alpha d}}\right)} \mu(\mathbf{x}) d\mathbf{z}$$

$$\le \int_{B\left(\mathbf{x}, w^{\frac{1}{\alpha d}}\right)} \frac{c}{V_d r^d} d\mathbf{z} = \frac{c}{r^d} w^{\frac{1}{\alpha}}.$$

Correspondingly, by substituting $u = \frac{c}{r^d} w^{\frac{1}{\alpha}}$, we obtain for any $u \le 2^d c$ that $u \ge F\left(\frac{u^\alpha r^{\alpha d}}{c^\alpha}\right)$ and thus $F^{-1}(u) \ge \frac{u^\alpha r^{\alpha d}}{c^\alpha}$. Note that for any $i \in [k_m]$, $\left\|\mathbf{x} - \mathbf{x}_{(i)}\right\|^\beta < (2r)^{\alpha d}$ so $W_{(i)}$ satisfies the condition that $W_{(i)} \le (2r)^{\alpha d}$. As such, using Lemma A.2 we obtain

$$\forall i \in [k_m], \qquad W_{(i)} \stackrel{d}{=} F^{-1}(U_{(i)}) \ge \frac{U_{(i)}^\alpha r^{\alpha d}}{c^\alpha}. \quad (10)$$

Now for $T_1$, we have from Eq. (10):

$$-T_1 \stackrel{d}{=} -\sum_{i=1}^{k_m} \frac{1}{F^{-1}(U_{(i)})^\beta} \ge -\frac{c^\alpha}{r^{\alpha d}} \sum_{i=1}^{k_m} \frac{1}{U_{(i)}^\alpha} \ge_{(1)} -\frac{2 \cdot 3^\alpha}{(1 - \alpha)} \frac{c^\alpha}{r^{\alpha d}} \cdot m^\alpha k_m^{1-\alpha}$$

$$\ge_{(2)} -\frac{2 \cdot 3^\alpha}{(1 - \alpha)} \frac{c^\alpha}{r^{\alpha d}} \cdot m^\alpha \left(\frac{3cm}{2}\right)^{1-\alpha} = -m \cdot \frac{2^\alpha \cdot 3}{(1 - \alpha) r^{\alpha d}} \cdot c,$$

where (1) holds by Lemma C.4 with probability at least $1 - \tilde{\mathcal{O}}_{k_m}\left(\frac{1}{k_m} + \frac{1}{k_m^{\frac{1-\alpha}{\alpha}}}\right) = \mathcal{O}_m\left(\frac{1}{m} + \frac{1}{m^{\frac{1-\alpha}{\alpha}}}\right)$ and (2) follows from Eq. (9).

*Bounding $T_2$:* Using the fact that for any $i \in [m]$, $\|\mathbf{x} - \mathbf{x}_i\| \leq \|\mathbf{x}\| + \|\mathbf{x}_i\| \leq R + r$, and the bound on $k_m$ from Eq. (9), we have for any $p < 0.49$ that

$$T_2 \geq \frac{(1-2p)(m-k_m)}{(R+r)^{\alpha d}} \geq \frac{(1-2p)}{(R+r)^{\alpha d}} \cdot \left( m - \frac{3}{4}m \right) > m \cdot \frac{1}{200(R+r)^{\alpha d}}.$$

*Bounding $T_3$:* From Lemma C.2 and Eq. (9), it holds with probability at least $1 - 2\exp\left(-\sqrt{\frac{m}{4}}\right)$ that

$$T_3 \leq \frac{(m-k_m)^{\frac{3}{4}}}{(2r)^{\alpha d}} \leq m^{\frac{3}{4}} \cdot \frac{1}{(2r)^{\alpha d}}.$$

*Putting it Together:* For any $\epsilon > 0$ there is some $m_0 \in \mathbb{N}$, such that for any $m > m_0$, $-T_3 \geq -m\epsilon$. So overall, we obtain that with probability at least $1 - \tilde{\mathcal{O}}_m\left( \frac{1}{m} + \frac{1}{m^{\frac{1-\alpha}{\alpha}}} \right)$,

$$\frac{1}{m} \sum_{i=1}^{m} \frac{y_i}{\|\mathbf{x} - \mathbf{x}_i\|^{\alpha d}} \geq \frac{1}{m}(-T_1 + T_2 - T_3) > -\frac{2^\alpha \cdot 3}{(1-\alpha)r^{\alpha d}} \cdot c + \frac{1}{200(R+r)^{\alpha d}} - \epsilon$$

$$\geq -\frac{6}{(1-\alpha)r^{\alpha d}} \cdot c + \frac{1}{400(R+r)^{\alpha d}},$$

where the last line follows by using that $\alpha < 1$, and by fixing some sufficiently small $\epsilon$. Finally, fixing some $c \leq \frac{(1-\alpha)r^{\alpha d}}{2400(R+r)^{\alpha d}} = \frac{1-\alpha}{2400\left(1+\frac{R}{r}\right)^{\alpha d}}$ suffices to ensure that this is positive, implying $\hat{h}_\beta(\mathbf{x}) = 1$.

**Lemma C.2.** *Under Setting C, let $\mathbf{x} \in B(\mathbf{0}, r)$ and $k_m := |\{i \in [m] \mid \|\mathbf{x}_i\| \leq r\}|$. It holds with probability at least $1 - 2\exp\left(-\sqrt{m - k_m}\right)$ that*

$$\left| \sum_{i:\|\mathbf{x}_i\| \geq 3r} \frac{y_i - 1 + 2p}{\|\mathbf{x} - \mathbf{x}_i\|^\beta} \right| \leq \frac{(m-k_m)^{\frac{3}{4}}}{(2r)^{\alpha d}}.$$

*Proof.* Let $\xi_i$ be the random variable representing a label flip, meaning that $\xi_i$ is 1 with probability $p$ and $-1$ with probability $1 - p$, and $y_i = f^*(\mathbf{x}_i)\xi_i$ by assumption. For any $\mathbf{x}_i$ with $\|\mathbf{x}_i\| \geq 3r$, it holds that $f^*(\mathbf{x}_i) = 1$, and that $\|\mathbf{x} - \mathbf{x}_i\| \geq \|\mathbf{x}_i\| - \|\mathbf{x}\| \geq 2r$, and thus $\frac{y_i}{\|\mathbf{x} - \mathbf{x}_i\|^{\alpha d}}$ are bounded as

$$\left| \frac{y_i}{\|\mathbf{x} - \mathbf{x}_i\|^{\alpha d}} \right| \leq \frac{1}{(2r)^{\alpha d}}.$$

We thus apply Hoeffding's Inequality (cf. 49, Theorem 2.2.6) yielding that for any $t \geq 0$

$$\Pr\left( \left| \sum_{i:\|\mathbf{x}_i\| \geq 3r} \frac{y_i}{\|\mathbf{x} - \mathbf{x}_i\|^{\alpha d}} - \sum_{i:\|\mathbf{x}_i\| \geq 3r} \frac{1-2p}{\|\mathbf{x} - \mathbf{x}_i\|^{\alpha d}} \right| \geq t \right) \leq 2\exp\left( -\frac{t^2 (2r)^{2\alpha d}}{2(m-k_m)} \right).$$

In particular, we have that with probability at least $1 - 2\exp\left(-\frac{1}{2}\sqrt{m - k_m}\right)$ that

$$\left| \sum_{i:\|\mathbf{x}_i\| \geq 3r} \frac{y_i}{\|\mathbf{x} - \mathbf{x}_i\|^{\alpha d}} - (1-2p) \sum_{i:\|\mathbf{x}_i\| \geq 3r} \frac{1}{\|\mathbf{x} - \mathbf{x}_i\|^{\alpha d}} \right| \leq \frac{(m-k_m)^{\frac{3}{4}}}{(2r)^{\alpha d}}.$$

$\square$

**Lemma C.3.** *Under setting C, let $k_m := |\{i : \|\mathbf{x}_i\| \leq r\}|$, then it holds with probability at least $1 - 2\exp(-\frac{c^2 m}{2})$ that*

$$\frac{cm}{2} \leq k_m \leq \frac{3cm}{2}.$$

*Proof.* We can rewrite $k_m = \sum_{i=1}^m B_i$ where $B_i = 1$ if $\|\mathbf{x}_i\| \leq r$ and 0 otherwise. Notice that each $B_i$ is a Bernoulli random variable with parameter $c$, i.e $B_i$ is 1 with probability $c$ and 0 with probability $1 - c$. So by Hoeffding's inequality (cf. 49, Theorem 2.2.6), we have for any $t \geq 0$ that

$$\Pr\left(\left|\sum_{i=1}^m B_i - cm\right| \geq t\right) \leq \exp\left(-\frac{2t^2}{m}\right).$$

Taking $t = \frac{cm}{2}$ concludes the proof. $\qquad\square$

**Lemma C.4.** *It holds for any* $k \leq m \in \mathbb{N}$, $0 < \alpha < 1$ *that with probability at least* $1 - \tilde{\mathcal{O}}_k\left(\frac{1}{k} + \frac{1}{k^{\frac{1-\alpha}{\alpha}}}\right)$,

$$\sum_{i=1}^k \frac{1}{U_{(i)}^\alpha} \leq \frac{2 \cdot 3^\alpha}{1 - \alpha}.$$

*Proof.* Fix some $n_0 \leq k$ which will be specified later. Using Lemma A.3, we can write

$$\sum_{i=1}^k \frac{1}{U_{(i)}^\alpha} \overset{d}{=} \left(\sum_{i=1}^m E_i\right)^\alpha \left(\sum_{i=1}^k \frac{1}{\left(\sum_{j=1}^i E_j\right)^\alpha}\right)$$

$$= \underbrace{\left(\sum_{i=1}^m E_i\right)^\alpha}_{:=T_1} \left(\underbrace{\sum_{i=1}^{n_0} \frac{1}{\left(\sum_{j=1}^i E_j\right)^\alpha}}_{:=T_2} + \underbrace{\sum_{i=n_0}^k \frac{1}{\left(\sum_{j=1}^i E_j\right)^\alpha}}_{:=T_3}\right). \tag{11}$$

By Lemma D.1, for some absolute constant $C > 0$ it holds with probability $1 - 2(1 + \frac{1}{C})\exp(-Cn_0)$ that for all $n \geq n_0$,

$$\frac{1}{2} \leq \sum_{i=1}^n E_i \leq \frac{3n}{2}. \tag{12}$$

Conditioned on this even occurring, we use this to bound both $T_1$ and $T_3$. For $T_1$, Eq. (12) directly implies that $T_1 \leq \left(\frac{3}{2}m\right)^\alpha$. For $T_3$, using both Eq. (12) as well as the integral test for convergence we obtain

$$T_3 \leq 2^\alpha \sum_{i=n_0}^k \frac{1}{i^\alpha} \leq 2^\alpha \int_{n_0-1}^k \frac{1}{i^\alpha} \leq 2^\alpha \frac{k^{1-\alpha} - (n_0-1)^{1-\alpha}}{1 - \alpha}.$$

It remains to bound $T_2$. By definition of an exponential random variable, for any $t \geq 0$ it holds for any $E_i$ with probability at least $\exp(-t)$ (which is $\geq 1 - t$) that $E_i \geq t$. So taking $t = \left(\frac{n_0}{k^{1-\alpha}}\right)^{\frac{1}{\alpha}}$, it holds with probability at least $1 - \left(\frac{n_0}{k^{1-\alpha}}\right)^{\frac{1}{\alpha}}$ that $E_1 \geq \left(\frac{n_0}{k^{1-\alpha}}\right)^{\frac{1}{\alpha}}$. As a result,

$$T_2 \leq n_0 \cdot \frac{1}{E_1^\alpha} \leq n_0 \cdot \frac{1}{\left(\frac{n_0}{k^{1-\alpha}}\right)} = k^{1-\alpha}. \tag{13}$$

To ensure that the probability that both Eq. (12) and Eq. (13) hold is sufficiently high, we take $n_0 = \max\left(\frac{1}{C}\log(k), 2\right)$. As such, we obtain that with probability at least $1 - \tilde{\mathcal{O}}\left(\frac{1}{k} + \frac{1}{k^{\frac{1-\alpha}{\alpha}}}\right)$ that Eq. (11) can be bounded as

$$\sum_{i=1}^k \frac{1}{U_{(i)}^\alpha} \overset{d}{=} T_1(T_2 + T_3) \leq \left(\frac{3}{2}m\right)^\alpha \left(2^\alpha \frac{k^{1-\alpha} - (n_0-1)^{1-\alpha}}{1-\alpha} + k^{1-\alpha}\right)$$

$$\leq \left(\frac{3}{2}m\right)^\alpha \cdot k^{1-\alpha}\left(\frac{2^\alpha}{1-\alpha} + 1\right) \leq \left(\frac{3}{2}m\right)^\alpha \cdot k^{1-\alpha} \cdot 2 \cdot \frac{2^\alpha}{1-\alpha}$$

$$\leq \frac{2 \cdot 3^\alpha}{1-\alpha} \cdot m^\alpha k^{1-\alpha}.$$

$\qquad\square$

# D   Auxiliary lemma

**Lemma D.1.** *Suppose $(E_i)_{i \in \mathbb{N}} \overset{iid}{\sim} \exp(1)$ are standard exponential random variables. Then there exists some absolute constant $C > 0$ such that:*

*1. For any $n \in \mathbb{N}$ it holds that*

$$\Pr\left(\frac{n}{2} \leq \sum_{i=1}^{n} E_i \leq \frac{3n}{2}\right) \geq 1 - 2\exp(-Cn).$$

*2. For any $n_0 \in \mathbb{N}$ it holds that*

$$\Pr\left(\bigcap_{n=n_0}^{\infty}\left[\frac{n}{2} \leq \sum_{i=1}^{n} E_i \leq \frac{3n}{2}\right]\right) \geq 1 - 2\left(1 + \frac{1}{C}\right)\exp(-Cn_0).$$

*Proof.* Denote by $\|\cdot\|_{\psi_1}$ the sub-exponential norm of a random vector (for a reminder of the definition, see for example Vershynin [49], Definition 2.7.5). Each $E_i$ satisfies for any $t > 0$, $\Pr(E_i \geq t) \leq \exp(-t)$ implying that $\|E_i\|_{\psi_1} = 1$. By Vershynin [50, Remark 5.18], this implies $\|E_i - 1\|_{\psi_1} \leq 2$. So Bernstein's inequality for sub exponential random variables [49, Corollary 2.8.3] states that there exists some absolute constant $C' > 0$ such that for any $t \geq 0$

$$\Pr\left(\left|\left(\frac{1}{n}\sum_{i=1}^{n} E_i\right) - 1\right| \geq t\right) \leq 2\exp\left(-C'\min\left(\frac{t^2}{4}, \frac{t}{2}\right)n\right).$$

Taking $t = \frac{1}{2}$ and taking $C := \frac{C'}{16}$ yields

$$\Pr\left(\left|\sum_{i=1}^{n} E_i - n\right| \geq \frac{n}{2}\right) \leq 2\exp(-Cn).$$

This proves the first statement. For the second statement, we union bound and apply the integral test for convergence, to get that

$$\Pr\left(\bigcup_{n=n_0}^{\infty}\left[\left|\sum_{i=1}^{n} E_i - n\right| \geq \frac{n}{2}\right]\right) \leq \sum_{n=n_0}^{\infty} \Pr\left(\left|\sum_{i=1}^{n} E_i - n\right| \geq \frac{n}{2}\right)$$

$$\leq 2\sum_{n=n_0}^{\infty} \exp(-Cn)$$

$$\leq 2\exp(-Cn_0) + 2\int_{n_0}^{\infty} \exp(-Cn_0)$$

$$\leq 2\exp(-Cn_0) + \frac{2}{C}\exp(-Cn_0).$$

$\square$

# E   Additional experiment

In this appendix, we provide an extension of the spherical data experiment discussed in the main paper, demonstrating the effect of noisy sampling on our results.

We repeated the experiment with data sampled from the sphere the sphere $\mathbb{S}^2 \subset \mathbb{R}^3$, given by Eq. (5). This time, after sampling from the sphere, we added Gaussian noise distributed as $\mathcal{N}(\mathbf{0}, \sigma^2 I_3)$ to each data point independently, and examined the effect of the noise variance $\sigma^2$ on the exponent $\beta$ achieving minimal test error (corresponding to the intrinsic dimension in our theory).

The results are presented in Figure 6, with $m = 2000$, $p = 0.04$ and various values of $\sigma^2$.

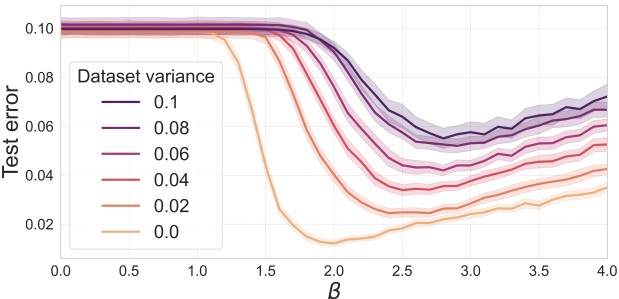

Figure 6: The classification error of $\hat{h}_\beta$ for varying values of $\beta$ and sampling noise $\sigma^2$. Best viewed in color.

As the noise increases, we see that this "best" $\beta$ *gradually* increases from 2 to 3, as the data indeed becomes fully dimensional for significant noise in the data-points. For example, with variance $0.04$ in each coordinate (equivalently, standard deviation $0.2$), the optimal $\beta$ is roughly $2.5$, which is notably still smaller than 3.

