# OpenReview forum: "Beyond Benign Overfitting in Nadaraya-Watson Interpolators"
_NeurIPS.cc/2025/Conference — NeurIPS 2025 poster_

### Official Review · Reviewer_eB3z · 2025-06-22

**Clarity:** 2
**Significance:** 2
**Originality:** 2
**Rating:** 4
**Confidence:** 3

**Summary:**

This paper studies the benign overfitting under the setting of Nadaraya-Watson(NW) estimators for binary classification with noise level $p \in (0,1/2)$ measured by the clean classification error.
In particular, the paper considers a special type of NW interpolator which is controlled by a hyperparameter $\beta$.
Previous work showed that if $\beta = d$, the interpolator has vanishing error as the sample size goes to infinity, which is benign overfitting.
This work focuses on the cases $\beta \neq d$ and shows that:  for $\beta > d$, the error is non-zero but bounded between $poly(p)$ and $O(p)$, corresponding to termpered overfitting; for $\beta < d$, the error is bounded below by some constant independent of $p$, corresponding to catastrophic overfitting.
Numerical simulations are also provided to validate the results.

**Questions:**

1. Can you compare the interpolating behaviors of the NW interpolator with other kind of (possibly benign) overfittings, such as the kernel interpolators in Mallinar et al. [7]  and http://arxiv.org/abs/2305.14077?

2. Can be discuss more on the intuition or insight of the parameter $\beta$? It seems that small $\beta$ leads to "underfitting" and large $\beta$ leads to "overfitting" (though it is already an interpolator).

3. Can the insights in this paper be applied to NW estimators for regression where there is a bandwidth parameter? What are the differences?

**Ethical Concerns:**

["NO or VERY MINOR ethics concerns only"]

**Final Justification:**

The authors' response clarifies the generality of their theory, particularly Theorem 4.1.
Now I tend to accept the paper because of the theoretical contributions.

**Limitations:**

yes

**Quality:**

3

**Strengths And Weaknesses:**

## Strengths
* This paper extends the understanding of benign overfitting to NW interpolators for binary classification.
* The non-monotonic phase change in the overfitting behaviors is novel and interesting.


## Weaknesses

* For the catastrophic overfitting, Theorem 4.1 is only stated for "there exist a density and a target function", but not for any density and target function, limiting the significance of the result.
* While the new phenomenon under the NW interpolator is discovered, more in-depth analysis and comparision can be provided to improve the understandings.

 I am willing increase my score if the weaknesses (particularly the first one) are properly addressed.

---

> ### Author Rebuttal · Authors · 2025-07-29
>
> We thank the reviewer for their time and constructive comments.
>
> - Theorem 4.1 statement: We thank the reviewer for this important comment, which we would like to thoroughly address. First, it is important to note that even the most trivial predictor can “accidentally” succeed at learning some target functions under certain densities. For example, consider the most trivial predictor which always predicts $0$ on points not in the training set: if the target function happens to be a constant zero, this predictor achieves perfect generalization. Of course, for almost any other target function, this predictor would exhibit terrible overfitting, and any sensible definition of catastrophic overfitting should be satisfied by such a predictor. We thus believe that the natural way to define catastrophic overfitting, as also done by all previous works, is in a worst-case manner, and therefore the statement of Theorem 4.1 is not a limitation of our analysis, but is rather a natural way to present a lower bound in this context. Nonetheless, we agree that we should better emphasize how our analysis applies to other constructions besides the specific counter-example in this theorem. Indeed, the analysis essentially shows that whenever a small region of constant probability mass is surrounded by the opposite label, it will be mislabeled by the predictor, incurring constant error. This is actually very generic, and applies to many ground truths $f^{\star}$. We will therefore revise the presentation to emphasize this point, and thank the reviewer for bringing this to our attention.
>
> - Comparison with other methods: There are by now many known results regarding the overfitting behavior of kernel interpolators, as partially summarized in the excellent paper by Mallinar et al. that you referenced. To the best of our knowledge, for kernel interpolators, there is no example of a single kernel provably exhibiting all three types of overfitting as we do here (even with a varying bandwidth). For example, as shown in [1], for any bandwidth, the RBF kernel is always worse than the constant-zero predictor, which implies catastrophic overfitting.
> Another important point is the difference in proof techniques. Known results for kernel interpolators typically rely on a spectral analysis. However, this often requires additional non-trivial assumptions (e.g. Gaussian universality). By contrast, our proof techniques are entirely different, based on characterizing the “locality” of the predictor. Following the reviewer’s comment, we will add this discussion to the final version of the paper.
>
> - Intuition regarding $\beta$: The hyperparameter $\beta$ influences “locality” of the predictor, meaning that as $\beta$ increases, the evaluation of the NW predictor at a new point will be influenced more by nearby points.
>
> - Extension for regression: Thank you for this excellent question. We suspect that the results should remain valid for regression, as the underlying idea behind the proofs should still hold. Specifically, if $\beta > d$ the predictor should still be “too local”, and if $\beta < d$, it should still be “too global”. Nevertheless, formalizing this would be an interesting future work.
>
> We hope our response addresses the reviewer’s comments, and will be happy to further clarify any point if needed. If there are any concerns remaining after this response that are preventing the reviewer from raising their score, please let us know so that we can address them.
>
> [1] Overfitting Behaviour of Gaussian Kernel Ridgeless Regression: Varying Bandwidth or Dimensionality, Medvedev et al., NeurIPS 2024

---

> > ### Comment · Reviewer_eB3z · 2025-08-01
> >
> > Thank you for clarifying my main concern. I would like to raise my score accordingly.

---

### Official Review · Reviewer_y616 · 2025-07-01

**Clarity:** 3
**Significance:** 3
**Originality:** 3
**Rating:** 4
**Confidence:** 4

**Summary:**

This paper analyzes overfitting of Nadaraya-Watson Interpolators, and identifies the benign overfitting regime encountered where $d=\beta$ (for dimension $d$ and bandwidth $\beta$) as a phase transition between so-called 'tempered' and 'catastrophic' regimes. These regimes are characterized in terms of the asymptotic classification error as a function of label noise.

**Questions:**

How do the estimators behave numerically at values of $p$ closer to 0.5?

ML problems frequently have high data dimension. How does varying the input dimension (both intrinsic and ambient) interact with your results. For GP regression problems, UQ quantities have been shown to be sensitive to data dimension, exhibiting monotonicity in some cases and double-descent in others. However these kernels do not include the one used here (since both the kernel and its gradient blow up at $\|x-y\|=0$). Any similarity or contrast in behavior of either $d$ or $d_\mbox{int}$ near $m$ could provide some insight here. Of course, there is also the risk that the results break down for high dimensional problems, so it should be confirmed that this is not the case. Similarly, data seldom has a clean intrinsic dimension, even if it’s just due to measurement or other noise. How does the addition of a small amount of noise perturbing the data from sub manifolds impact your results?

Is there any reason that there is inconsistency between the requirement that $p\int(0,\frac{1}{2})$ and $p\in(0,0.49)$ besides convenience? In my opinion, consistency here, and statement of the dependence away from $\frac{1}{2}$ would be preferable.

**Ethical Concerns:**

["NO or VERY MINOR ethics concerns only"]

**Final Justification:**

This is a good paper, and I think it should be accepted. Being primarily a theory paper, the experimental section was weak, but the authors have provided some additional experiments that I believe strengthen the work.

**Limitations:**

Limitations of the work are not explicitly addressed.

**Paper Formatting Concerns:**

-

**Quality:**

3

**Strengths And Weaknesses:**

The paper is generally well-written and the proofs appear to be correct.
The numerics section of the paper only considers the cases where $d=1$, and $d=3$ with intrinsic dimension of $d_\mbox{int}=2$.
Similarly, the behavior is only treated numerically for small values of $p$.
The claims switch from $p\in(0,\frac{1}{2})$ to $p\in(0,0.49)$ which appears to be purely for convenience (in the calculation of $T_2$ in the proof of Theorem 4.1). Restatement of the theorems in terms of $(1-2p)$ may be illustrative.

I think this is good work and would like to be able to raise my score if the paper is improved.

---

> ### Author Rebuttal · Authors · 2025-07-29
>
> We thank the reviewer for their time and constructive comments. Below, we address all of the comments raised by the reviewer:
>
> - Value of the noise parameter $p$: Indeed, as the reviewer correctly noted, the choice of stating the claims for $p\in (0, 1/2)$ or $p \in (0, 0.49)$ is purely for convenience, as the term $1-2p$ appears in the proof of Theorem 4.1. As $p$ approaches $0.5$, the training labels approach pure noise, regardless of $f^{\star}$. Therefore, previous works focus on the regimes in which $p$ is small, since that is when tempered overfitting is far less harmful than catastrophic, which is of key interest. We emphasize that the ability to cover the entire noise interval is actually an advantage of our analysis, rather than a drawback. We agree that consistency in the statement of the theorems would improve the presentation of the paper, and we will make the necessary adjustments. Thank you for this suggestion!
> - Dimension of the data: An important observation is that the NW predictor depends entirely on the distances between points, and as such, the results depend on the intrinsic dimension of the data, and not the ambient dimension (as exemplified in Figure 4 in the paper). The reviewer raised a very interesting question regarding noise, which can influence this ambient dimension. Yet, since the predictor only depends on the distances between points, the predictor is robust to sufficiently small perturbations, which would hardly affect the distances. Indeed, we have new simulations showing this phenomenon, which we will add to the final version of the paper (note that we cannot add them here due to the new NeurIPS policy).
> Lastly, regarding the interaction between the dimension and number of samples, the proofs in the paper hold for any dimension, though of course, the constants and the rates may depend on the dimension. This is inevitable, as a larger dimension may lead to slower convergence, as is to be expected for non-parametric learning rules. Please see also our 3rd item in response to reviewer bJoP on this point.
>
> We hope our response addresses the reviewer’s comments, and will be happy to further clarify any point if needed. If there are any concerns remaining after this response that are preventing the reviewer from raising their score, please let us know so that we can address them.

---

> > ### Comment · Reviewer_y616 · 2025-08-01
> > **Additional experiments**
> >
> > Thanks for running those experiments investigating the effects of noise on the ambient dimensions. While I understand that I cannot see an image displaying the results, I would like to see a description and tabulated results from your experiments, including the sensitivity to varying levels of noise. While it should be expected that this effect is reasonably robust to small perturbations, in my view it would be interesting to demonstrate how this robustness decays as the noise increases. A throwaway comment that it's not very sensitive if you don't inject much noise is not enough.
> >
> > Similarly, I understand that the kernels used only depend on the distance for fixed dataset and ambient dimension. However, the behavior of related families of kernel matrices are well-known to depend on the dimension of the underlying data, relative to the number of data points [1]. This has downstream effects on the level of the marginal likelihood for GP regression, where it has been shown that adding additional dimensions of pure noise has a regularizing effect [2]. The interplay between this phenomena and the effective dimension of the dataset, as well as the relationship between the dataset size and ambient/effective dimension are not trivial, and can impact model performance.
> >
> > Machine learning problems typically contain far more than 1-3 dimensions, and datasets can be noisy. Testing only on two very simplistic cases is probably not enough for this outlet, particularly if reviewers have requested to see the effects of other factors. I am happy to change my assessment of your work, but in my opinion these effects are important and their investigation would improve your work. As it stands, simply dismissing them is not convincing. It is unfortunate that images cannot be displayed during this period, but this just means that results have to be conveyed in alternative ways.
> >
> > [1] El Karoui, N. The spectrum of kernel random matrices. The Annals of Statistics, 38(1):1–50, 2010.
> > [2] Hodgkinson, L. et al. Monotonicity and double descent in uncertainty estimation with Gaussian processes. ICML (pp. 13085–13117), 2023.

---

> ### Author Response · Authors · 2025-08-03
> **Additional Experiments**
>
> We would like to thank the reviewer again for their rapid response and their continued engagement. We will now describe 3 new experiments aimed at addressing the reviewer’s comments. All of the raw data needed to fully reproduce the described figures is provided at the end (due to character limit constraints, some of the raw data will be provided in the next comment). We believe these experiments strongly demonstrate the validity of our findings in natural scenarios in which, as the reviewer noted, data points are noisy and reside in a substantially higher dimension.
>
> - **MNIST data**: In this experiment, we took images of 0s and 1s from the MNIST dataset, flipped each label with probability p, and considered a NW kernel classifier which performs binary-classification of these classes based on the noisy training data. Our results are consistent with our theory. In particular, for small exponent $\beta$ the test error is roughly 0.2 across all noise levels, which then decreases as $\beta$ grows until roughly $\beta \sim 8-10$. After that, the test error gradually increases at a mild rate as beta grows. The increase in test error is extremely mild for lower noise rates, and steeper as the label noise is higher, very similarly to the left figures 3 and 4 in our paper. It is interesting to note that although MNIST images are of “extrinsic” dimension 784(=28x28), the minimum test error across all noise levels is around $\beta \sim 8$, which matches the estimate of the intrinsic dimension of MNIST according to the paper “The intrinsic dimension of images and its impact on learning” by Pope et al. ICLR 2021.
>
> - **The effect of sample size in MNIST**: In this experiment, we analyze the effect of varying the training set size in a high-dimensional setting given by MNIST. Specifically, we fix the noise level at p=0.1 and for different training set sizes, $m=50, 100, 500, 2000, 10000$, we plot the binary-classification test error for various values of $\beta$. For every $m$ that we plotted, the predictor undergoes the catastrophic-benign-tempered phenomenon described by our theory. The difference between different values of training sizes $m$ is very similar to the right figures 3 and 4 in the paper, where the underlying shape of the plot does not change, but it becomes more pronounced and clearer as the number of samples increases. We emphasize that unlike the experiments in the paper, the dimension of the data here (d=784) is larger than some of the values of $m$ that we plot.
>
> - **Noisy synthetic data**: To further demonstrate the effect of noisy sampling on our results, we repeated the experiment described in our paper of data on the 2-dimensional sphere embedded in 3d. This time, after sampling from the sphere, we added Gaussian noise to each data point independently, and examined the effect of the noise variance on the exponent $\beta$ achieving minimal test error (corresponding to the intrinsic dimension in our theory). As the noise increases, we see that this “best” $\beta$ *gradually* increases from 2 to 3, as the data indeed becomes fully dimensional for significant noise in the data-points. For example, with standard deviation 0.02 in each coordinate, the optimal $\beta$ is 2.3, and for noise as large as having standard deviation 0.1 in each coordinate, the optimal $\beta$ grows to 2.7 (still < 3).
>
> We hope these additions address the reviewer’s comments, and will be happy to further clarify any point if needed.
>
> **Raw data #1:** for different values of p, we report the test error for beta in [1,2,...,40] (by this order):
>
> p=0.01: [0.176501, 0.077920, 0.051726, 0.034704, 0.027376, 0.020473, 0.013995, 0.010213, 0.008794, 0.007991, 0.007518, 0.007329, 0.006288, 0.006383, 0.006478, 0.005437, 0.005768, 0.005437, 0.005674, 0.005863, 0.005816, 0.007896, 0.007612, 0.007329, 0.007423, 0.008085, 0.008038, 0.009456, 0.007801, 0.007943, 0.008180, 0.008511, 0.009220, 0.009173, 0.008747, 0.009362, 0.008747, 0.008558, 0.008558, 0.009409]
>
> p=0.05: [0.165816, 0.077494, 0.050686, 0.035083, 0.027518, 0.020709, 0.014374, 0.010449, 0.010355, 0.010307, 0.010969, 0.011395, 0.014468, 0.014610, 0.015603, 0.018251, 0.021087, 0.024870, 0.025485, 0.025768, 0.027139, 0.030165, 0.032009, 0.032813, 0.033522, 0.034043, 0.036312, 0.034799, 0.035556, 0.037589, 0.040426, 0.038061, 0.039054, 0.041655, 0.043404, 0.042506, 0.041135, 0.041560, 0.043262, 0.043499]
>
> p=0.1: [0.173853, 0.078723, 0.051158, 0.035745, 0.027470, 0.020662, 0.015603, 0.011584, 0.012151, 0.014988, 0.017778, 0.023641, 0.024019, 0.029362, 0.033617, 0.038818, 0.042742, 0.048369, 0.052482, 0.057258, 0.058960, 0.060804, 0.065957, 0.064161, 0.065201, 0.072671, 0.071631, 0.074137, 0.074090, 0.074137, 0.077683, 0.080946, 0.085768, 0.079338, 0.088227, 0.082364, 0.084161, 0.085957, 0.083452, 0.089078]

---

> > ### Comment · Reviewer_y616 · 2025-08-03
> > **Additional experiments**
> >
> > Thanks for running those experiments, they clarify the main issues that I had. I really like the MNIST example. Matching the intrinsic dimension found by your method to previous work is really cool, and in my opinion validates your work. I hope that you include this and the other experiments in the final version of the paper. I have raised my score accordingly.

---

> ### Author Response · Authors · 2025-08-03
> **Additional Experiments (continued)**
>
> p=0.2: [0.170544, 0.084255, 0.049220, 0.034894, 0.027376, 0.021986, 0.017447, 0.016501, 0.022506, 0.029409, 0.041797, 0.049504, 0.062884, 0.073522, 0.079102, 0.093522, 0.103924, 0.111111, 0.120284, 0.130922, 0.137636, 0.134232, 0.142742, 0.149314, 0.148085, 0.159622, 0.158771, 0.158818, 0.163262, 0.161702, 0.169409, 0.169645, 0.169929, 0.176785, 0.171017, 0.175745, 0.184113, 0.181891, 0.175839, 0.179385]
>
> **Raw data #2:** for different values of training set size m, we report the test error for beta in [1,2,...,40] (by this order):
>
> m = 50: [0.24158, 0.15991, 0.12369, 0.08288, 0.07031, 0.07281, 0.06986, 0.06979, 0.06527, 0.07982, 0.07312, 0.07743, 0.10461, 0.10214, 0.08912, 0.095, 0.09937, 0.08544, 0.08569, 0.09959, 0.10461, 0.1014, 0.10304, 0.09001, 0.08196, 0.09106, 0.09706, 0.10603, 0.11311, 0.10073, 0.09435, 0.10225, 0.10024, 0.09331, 0.10461, 0.096, 0.11139, 0.09497, 0.09192, 0.12259]
>
> m = 100: [0.1843, 0.1504, 0.06483, 0.06156, 0.05134, 0.05068, 0.06227, 0.05275, 0.0588, 0.0672, 0.06795, 0.06897, 0.06861, 0.07655, 0.08139, 0.07116, 0.08422, 0.08651, 0.08659, 0.08565, 0.09302, 0.08626, 0.09039, 0.09072, 0.09179, 0.09117, 0.08974, 0.09515, 0.09108, 0.09108, 0.09672, 0.10749, 0.10072, 0.10348, 0.09191, 0.09139, 0.10408, 0.09684, 0.10074, 0.10019]
>
> m = 500: [0.19555, 0.09268, 0.06045, 0.03926, 0.03301, 0.02845, 0.02707, 0.03001, 0.03531, 0.03882, 0.04384, 0.05052, 0.05635, 0.05969, 0.05998, 0.06296, 0.06963, 0.07355, 0.07126, 0.07926, 0.08004, 0.07835, 0.08281, 0.07946, 0.08355, 0.08527, 0.08827, 0.08753, 0.08736, 0.0914, 0.08778, 0.08904, 0.08893, 0.09106, 0.09093, 0.09434, 0.09965, 0.09145, 0.09143, 0.0914]
>
> m = 2000: [0.17536, 0.08766, 0.04944, 0.03848, 0.02704, 0.02266, 0.01861, 0.01901, 0.02201, 0.02566, 0.03051, 0.03504, 0.04033, 0.04522, 0.04958, 0.05415, 0.05845, 0.06087, 0.06436, 0.06805, 0.06787, 0.07159, 0.07415, 0.0751, 0.07779, 0.07855, 0.08267, 0.08087, 0.08283, 0.08389, 0.08488, 0.08552, 0.08861, 0.0873, 0.08989, 0.08939, 0.09287, 0.08996, 0.08891, 0.0926]
>
> m = 10000: [0.16919, 0.07766, 0.05123, 0.03544, 0.02762, 0.02066, 0.015, 0.01189, 0.01268, 0.01539, 0.01944, 0.0231, 0.02679, 0.03174, 0.03673, 0.04199, 0.04513, 0.04803, 0.05144, 0.05546, 0.0597, 0.06145, 0.06364, 0.06624, 0.06847, 0.07245, 0.07391, 0.07482, 0.07677, 0.07766, 0.07944, 0.08014, 0.08122, 0.08213, 0.0831, 0.08354, 0.08655, 0.08744, 0.08638, 0.08815]
>
>
>
> **Raw data #3:** for different values of standard deviation (per coordinate) sigma, we report the test error for beta in [0, 0.1, 0.2, … 4]: (we considered 1K training points, averaged across 50 runs)
>
> $\sigma$=0: [0.09962, 0.09962, 0.09962, 0.09962, 0.09962, 0.09962, 0.09962, 0.09962, 0.09960, 0.09954, 0.09926, 0.09810, 0.09324, 0.08514, 0.06758, 0.05262, 0.04154, 0.03350, 0.02960, 0.02844, 0.02894, 0.03270, 0.03390, 0.03770, 0.04058, 0.04530, 0.04860, 0.05168, 0.05404, 0.05904, 0.05818, 0.06386, 0.06344, 0.06642, 0.06908, 0.07482, 0.07128, 0.07886, 0.07652, 0.07798, 0.08104]
>
> $\sigma$=0.02: [0.09990, 0.09990, 0.09990, 0.09990, 0.09990, 0.09990, 0.09990, 0.09990, 0.09990, 0.09990, 0.09990, 0.09990, 0.09994, 0.09980, 0.09874, 0.09574, 0.08622, 0.07466, 0.06518, 0.05338, 0.04610, 0.04218, 0.04016, 0.03852, 0.03866, 0.03904, 0.04074, 0.04360, 0.04772, 0.04744, 0.05270, 0.05248, 0.05734, 0.05880, 0.06230, 0.06362, 0.06754, 0.06886, 0.07316, 0.07616, 0.07978]
>
> $\sigma$=0.04: [0.10012, 0.10012, 0.10012, 0.10012, 0.10012, 0.10012, 0.10012, 0.10012, 0.10012, 0.10012, 0.10012, 0.10012, 0.10012, 0.10006, 0.09982, 0.09884, 0.09518, 0.08976, 0.08120, 0.07310, 0.06282, 0.05688, 0.05414, 0.04956, 0.04908, 0.04714, 0.05180, 0.05172, 0.05190, 0.05158, 0.05474, 0.05882, 0.06626, 0.06502, 0.06960, 0.06828, 0.07336, 0.07658, 0.07608, 0.08114, 0.08476]
>
> $\sigma$=0.06: [0.09888, 0.09888, 0.09888, 0.09888, 0.09888, 0.09888, 0.09888, 0.09888, 0.09888, 0.09888, 0.09888, 0.09886, 0.09886, 0.09874, 0.09870, 0.09840, 0.09708, 0.09444, 0.08902, 0.08200, 0.07548, 0.06948, 0.06406, 0.06084, 0.05858, 0.06036, 0.05760, 0.05886, 0.06130, 0.06202, 0.06346, 0.06864, 0.07196, 0.07012, 0.07650, 0.07858, 0.08220, 0.08304, 0.08382, 0.09192, 0.09274]
>
> $\sigma$=0.08: [0.10038, 0.10038, 0.10038, 0.10038, 0.10038, 0.10038, 0.10038, 0.10038, 0.10038, 0.10038, 0.10038, 0.10036, 0.10036, 0.10026, 0.10034, 0.10006, 0.09954, 0.09816, 0.09448, 0.09146, 0.08678, 0.08314, 0.07396, 0.07278, 0.06802, 0.06812, 0.06678, 0.06630, 0.06548, 0.06686, 0.06752, 0.07182, 0.07368, 0.07996, 0.07760, 0.08324, 0.08578, 0.08834, 0.09282, 0.09390, 0.09546]
>
> $\sigma$=0.1: [0.09942, 0.09942, 0.09942, 0.09942, 0.09942, 0.09942, 0.09942, 0.09942, 0.09942, 0.09942, 0.09942, 0.09942, 0.09942, 0.09940, 0.09932, 0.09924, 0.09908, 0.09796, 0.09660, 0.09430, 0.08936, 0.08604, 0.08424, 0.07834, 0.07412, 0.07358, 0.07360, 0.06870, 0.07416, 0.07414, 0.07812, 0.07918, 0.08124, 0.08666, 0.08694, 0.08736, 0.09186, 0.09530, 0.09706, 0.10012, 0.10110]

---

### Official Review · Reviewer_bJoP · 2025-07-03

**Clarity:** 4
**Significance:** 4
**Originality:** 4
**Rating:** 4
**Confidence:** 2

**Summary:**

The paper analyzes the generalization performance of the NW interpolator. The results show that if $\beta>d$, there exists tempered overfitting, meaning the excess risk has the same order as the noise rate. If $\beta<d$, there exists catastrophic overfitting, meaning that the excess risk may be $\Theta(1)$ large.

**Questions:**

Is it possible to extend your result to multi-class classification? What will the result look like?

Is it possible to extend your result to a setting in which the noise rate is data dependent? (p is a function of x)

What is the convergence rate of the NW interpolator? Can you express the excess risk as a function of m? Is it in general better than using a neural network to solve the classification problem?

**Ethical Concerns:**

["NO or VERY MINOR ethics concerns only"]

**Final Justification:**

This is a very good paper and the results are interesting. I recommend accepting this paper.

**Limitations:**

yes

**Paper Formatting Concerns:**

No such issue

**Quality:**

4

**Strengths And Weaknesses:**

The introduction and the main results are stated very clearly.

Proof sketches and the intuitions behind proofs are explained in detail.

Experiments are presented to justify the theories.

The solved problem is technical and contributive.

---

> ### Author Rebuttal · Authors · 2025-07-29
>
> We thank the reviewer for their time, and are encouraged by their appreciation of our contribution. Below, we address the reviewer’s questions:
> - Extensions to the multi-class setting are a very interesting direction, which are generally under-studied in the modern overfitting context (see [1] for a notable exception). We would like to remark that for kernel rules, there is generally not a single canonical way to extend the predictor itself to the multi-class setting. If viewing the outputs as vector valued and taking the argmax, we do expect the results to generalize. In any case, this is certainly an interesting direction for future research.
> - It is very likely that our results extend to the case in which the noise rate is data-dependent. Indeed, since our tempered analysis is highly local, it is likely that p can be replaced by a localized quantity such as $\int p(x) d\mu(x)$, namely the averaged noise rate with respect to the underlying distribution. As to the catastrophic case, the results should readily hold for data-dependent p as long as it is bounded away from 0 and 1/2, which would not actually alter the analysis. We thank the reviewer for this excellent point, and we will remark about this in the final version of the paper.
> - Convergence rates of NW interpolators and related non-parametric rules are extensively studied in the literature, and such rates generally depend on the smoothness properties of the underlying $f^{\star}$, see e.g. the books [2,3]. NW interpolators are one of the most well-studied learning rules in statistics for decades, and the question of comparing whether they are preferable or not to neural networks is beyond the scope of this work. In our work, we chose not to make any such quantitative assumptions on  $f^{\star}$ for the results to be as generally applicable as possible, and therefore did not go into analyzing the corresponding rates. We remark that indeed using these existing techniques, it is not difficult to establish the corresponding rates for smoothness classes.
>
> We hope our response addresses the reviewer’s comments, and will be happy to further clarify any point if needed.
>
> References:
>
> [1] Benign Overfitting in Multiclass Classification: All Roads Lead to Interpolation, Wang et al. NeurIPS 2021
>
> [2] Introduction to Nonparametric Estimation, by Tsybakov
>
> [3] A probabilistic theory of pattern recognition, by Devroye, Gyorfi and Lugosi

---

> ### Comment · Reviewer_bJoP · 2025-08-08
>
> In case I am flagged, I need to make some comments. The questions that I asked in my original review were quite challenging, so I didn't expect the authors to address them. Overall, this is an interesting paper, so I will maintain my original score.

---

### Official Review · Reviewer_AL49 · 2025-07-03

**Clarity:** 3
**Significance:** 2
**Originality:** 3
**Rating:** 5
**Confidence:** 3

**Summary:**

The authors study different generalization regimes using a simple well-defined interpolating model, the Nadaray-Watson kernel estimator for binary classification. They show that even this simple model can exhibit complex generalization behavior and that the progression from catastrophic, to benign, to tempered overfitting occurs, and - in this case - can be controlled by a single hyperparameter: the exponent of the kernel denominator (beta). The main contribution of the paper is theoretical, with proofs provided of the role of beta in the progression from one regime to another. This is supplemented with experimental verification using synthetic data.

**Questions:**

Please discuss the significance of the findings.

**Ethical Concerns:**

["NO or VERY MINOR ethics concerns only"]

**Final Justification:**

Solid technical paper. Novel contribution to relevant field. Significance for the broader field is somewhat borderline, but I believe it to be over the threshold.

**Limitations:**

yes

**Quality:**

4

**Strengths And Weaknesses:**

Strengths:
- A very specific theoretical question is precisely answered.
- Claims are clear and well supported.
- Theoretical analysis is similarly well presented and an effort is made to make the proofs accessible.
- Work is carefully developed and the methodology used is sound.
- Results are interesting and novel.
- The discussion around explicit vs implicit dimension and the implications of over- or underestimating dimensionality contributes to the main result.

Weaknesses:
- I am uncertain of the significance of the main analysis.

Other comments:
- Line 96 – introduction to sections needs an update.
- I did not review the proofs in the supplementary material.

---

> ### Author Rebuttal · Authors · 2025-07-29
>
> We thank the reviewer for their time, and are encouraged by their appreciation of our contributions. We also thank them for spotting the need for updating the section numbering in Line 96, which we will incorporate into the final version.
>
> Addressing the reviewer’s question, we believe the significance of our results is two-fold:
> - Overfitting regimes of interpolating predictors are a widely studied topic recently. However, we are not aware of any single interpolator that provably exhibits the full range of catastrophic-tempered-benign overfitting behaviors, as we show here. Thus, we identify a simple and classical predictor whose analysis allows us to understand the full spectrum of overfitting behaviors, and the factors that underlie them (such as the effect of locality).
> - By revisiting the analysis and extending it to inconsistent regimes not previously covered, we identify an interesting dependence on the *intrinsic* dimension, which is typically unknown. This phenomenon, also backed up by numerical evidence, has an implication on the optimal choice of the hyperparameter $\beta$ in practice, as we further discuss in the paper.
>
> We hope our response addresses the reviewer’s comments, and will be happy to further clarify any point if needed.

---

> ### Comment · Reviewer_AL49 · 2025-08-04
>
> While these aspects are touched on in your paper, they do not come across as clearly in your paper as here. (See your Discussion, for example.) I suggest you put more emphasis on these aspects - they better highlight the significance of the work to your reader. Thanks for the answer - I understand better.
>
> I gave this paper a good score during my first review, and see no red flags in any of the other reviews. My main concern is how significant these findings are for the larger field, but as a novel contribution to a relevant topic, I believe it to be over the threshold.

---

### Decision · Program_Chairs · 2025-09-17

**Decision:**

Accept (poster)

**Comment:**

The paper provides the overall overfitting profiles for the Nadaraya-Watson Interpolators. The key is the selection of $\beta$ in the inverse distance weighting Kernel $K(x,y)=1/||x-y||^\beta$. (Since the inverse distance weighting interpolates the data. ) When $\beta$ is smaller than $d$, the classifier would seek data apart from the origin, thus making catastrophic overfitting results. When $\beta$ is larger than $d$, the estimator is similar to a $k$-nearest neighbor estimator with finite $k$. The proof sketch is clean and easy to follow, providing a comprehensive understanding of the Nadaraya-Watson Interpolators. Thus, I suggest acceptance.